# Object knowledge representation in the human visual cortex requires a connection with the language system

**Bo Liu[1,2,3,4], Xiaosha Wang[1], Xiaoying Wang[1], Yan Li[2,3,4], Yang Han[3,5], Jiahui Lu[1], Hui Zhang[2,3,4], Xiaochun Wang[2,3,4]\*, Yanchao Bi** [1,6,7,8,9]\*

**1** State Key Laboratory of Cognitive Neuroscience and Learning & IDG/McGovern Institute for Brain Research, Beijing Normal University, Beijing, China, **2** Department of Radiology, First Hospital of Shanxi Medical University, Taiyuan, Shanxi, China, **3** College of Medical Imaging, Shanxi Medical University, Taiyuan, Shanxi, China, **4** Shanxi Key Laboratory of Intelligent Imaging, First Hospital of Shanxi Medical University, Taiyuan, Shanxi, China, **5** Xi'an Key Laboratory of Metabolic Disease Imaging, Xi'an No.3 Hospital, Affiliated Hospital of Northwest University, Xi'an, China, **6** School of Psychological and Cognitive Sciences and Beijing Key Laboratory of Behavior and Mental Health, Peking University, Beijing, China, **7** IDG/McGovern Institute for Brain Research, Peking University, Beijing, China, **8** Institute for Artificial Intelligence, Peking University, Beijing, China, **9** Key Laboratory of Machine Perception (Ministry of Education), Peking University, Beijing, China

\* wangxiaochun@sydyy.com (XW); ybi@pku.edu.cn (YB)

**Academic editor:** Huan Luo, Peking University, CHINA

## Abstract

How world knowledge is stored in the human brain is a central question in cognitive neuroscience. Object knowledge effects have been commonly observed in higher-order sensory association cortices, with the role of language being highly debated. Using object color as a test case, we investigated whether communication with the language system plays a necessary role in knowledge neural representation in the visual cortex and corresponding behaviors, combining diffusion imaging (measuring white-matter structural integrity), functional MRI (fMRI; measuring functional neural representation of knowledge), and neuropsychological assessments (measuring behavioral integrity) in a group of patients who suffered from stroke ($N = 33$; 18 with left-hemisphere lesions, 11 with right-hemisphere lesions, and 4 with bilateral lesions). The structural integrity loss of the white-matter connection between the anterior temporal language region and the ventral visual cortex had a significant effect on the neural representation strength of object color knowledge in the ventral visual cortex and on object color knowledge behavior across modalities. These contributions could not be explained by the potential effects of the early visual perception pathway or potential confounding brain or cognitive variables. Our experiments reveal the contribution of the vision-language connection in the ventral occipitotemporal cortex (VOTC) object knowledge neural representation and object knowledge behaviors, highlighting the significance of the language-sensory system interface.

**Data availability statement:** All the data necessary to reproduce Figs 2–4, S3–S5 and Tables 1, S4–S6 are provided in S1 Data. The reconstructed WM tract masks, the VOTC masks, and the MATLAB codes for RSA and probabilistic tractography analysis are publicly available at Zenodo (https://zenodo.org/records/15070254; DOI: 10.5281/zenodo.15070254). The neuroimaging data supporting the current study are not shared openly due to ethical constraints, but are available from the Ethics Committee of the First Hospital of Shanxi Medical University (phone: +86 351 4639242) for researchers who meet the criteria for access to confidential data.

**Funding:** This work was supported by the STI2030-Major Project (https://en.most.gov.cn/, 2021ZD0204100 (2021ZD0204104) to Y.B.), the National Natural Science Foundation of China (https://www.nsfc.gov.cn/english/site_1/index.html, 31925020 and 82021004 to Y.B., 32171052 to X.S.W., 32071050 to X.Y.W.), and the Fundamental Research Funds for the Central Universities (http://en.moe.gov.cn/, to Y.B.). The funders had no role in study design, data collection and analysis, decision to publish, or preparation of the manuscript.

**Competing interests:** The authors have declared that no competing interests exist.

**Abbreviations :** ATL, anterior temporal lobe; CSF, cerebrospinal fluid; EPI, echo-planar imaging; FA, fractional anisotropy; FDR, false discovery rate; FLAIR, fluid-attenuated inversion recovery; FLIRT, FMRIB's linear image registration tool; fMRI, functional MRI; FMRIB, Functional Magnetic Resonance Imaging of the Brain; FNIRT, FMRIB's non-linear image registration tool; FOV, field of view; GLM, general linear model; HARDI, high angular resolution diffusion imaging; IFOF, inferior fronto-occipital fasciculus; ILF, inferior longitudinal fasciculus JHU, Johns Hopkins University; LAG, left angular gyrus; LdlATL, left dorsolateral anterior temporal lobe; LIFGorb, left inferior frontal gyrus, orbital part; LMFG, left middle frontal gyrus; LpMTG, left posterior middle temporal gyrus; MMSE, Mini-Mental State Examination; MNI, Montreal Neurological Institute; OP, occipital pole; RDM, representational dissimilarity matrix; RF, radio frequency; ROI, region of interest; RSA, representation similarity analysis; SD, standard deviation; SPACE, Sampling Perfection with Application optimized Contrast using different flip angle Evolution; TE, echo time; TR, repetition time; VOTC, ventral occipitotemporal cortex; WM, white-matter.

## Introduction

The human brain stores rich object knowledge that supports our interactions with objects through various sensorimotor modalities. How such memory is stored in the brain is one of the core questions in cognitive neuroscience. Decades of brain imaging and neuropsychological studies have converged on revealing that knowledge about specific object attributes is grounded in higher-order sensory association cortices, derived from perceptual experiences (see reviews in [1–3]), which are presumably then bound together to form higher-order object concepts [4,5]. One line of the foundational evidence comes from object color knowledge. Retrieving knowledge about object color, when presented with a name or a grayscale object image (e.g., red for rose) or when processing objects with salient diagnostic colors relative to other objects (e.g., rose versus chair), activates regions in the ventral occipitotemporal cortex (VOTC) that also engage in perceiving colors [6–11]. The classical cases with object color knowledge deficits [12–14] had lesions spanning the ventral temporal cortex, including, although not confined to, the occipitotemporal cortex (i.e., extending to the ventral medial temporal pole regions). Thus, the current consensus is that the VOTC is the neural correlate supporting color knowledge, related to visual color perception.

Whether object perception functionality in the VOTC, and by extension, the perception-derived knowledge representation, is driven only by visual/sensory properties or modulated by various high-order computations, especially language, remains vigorously debated [e.g., 15–19]. For example, while Wang and colleagues [19] showed that large visual computation models that incorporate language supervision yield additional explanatory power of the human VOTC functionality over pure vision models, other researchers have shown that such advantages might be due to differences other than language involvements across models ([18]; see also discussion in [19]), and pure vision models can well explain many seemingly high order visual functionalities in the VOTC [e.g., 15–17]. A key challenge is that in a typical system, information structures derived from various levels of perception and language are difficult to disentangle (*cat* and *dog* are related both in visual and language spaces, and in tactile and auditory spaces).

In the case of object color, a line of recent studies with congenitally blind individuals (i.e., resulting in full deprivation of color sensory experiences) has identified object color knowledge representation in the dorsal anterior temporal lobe (ATL), in addition to the VOTC. This representation is thus likely derived from language experience ([20–23]; see [24] for review); the dorsal ATL cluster is intrinsically and functionally connected to the language system and with the VOTC in healthy controls [22]. This anterior non-sensory color knowledge representation thus constitutes a strong candidate that may support normal object color knowledge in individuals with congenital blindness, or patients with cortical blindness from bilateral V1 lesions or with cerebral achromatopsia [22,25,26]. These findings raise two questions: (1) Is the VOTC's object knowledge representation affected by such connections with the anterior, language-derived, representation? (2) Do such connections with the language system contribute to supporting object color knowledge behavior? In other words, is the neural representation in the VOTC sufficient to "know" that a banana shape should be yellow in color without connection with anterior language-derived regions?

Suggestive evidence that VOTC functionality alone may be insufficient to support object color knowledge and thus may depend on its communication with higher-order regions comes from studies of patients with semantic dementia. Rogers and colleagues [27] reported that semantic dementia patients, who had diffuse atrophy predominantly in the ATL, exhibited severe deficits in object color knowledge, even in non-verbal visual tasks, despite having relatively preserved sensory association cortices and primary perceptual processing (assumed to reflect intact VOTC functionality). A recent study further revealed that object color knowledge deficits in semantic dementia patients were related to decreased structural connections between the left fusiform gyrus (spanning from the anterior to posterior ventral temporal lobe) and the early visual cortex (calcarine) [28]. Given that brain atrophy in semantic dementia patients may affect the functionality of surrounding or remote areas [29–31], it is unclear what aspects of this connection in relation to the VOTC account for the object color knowledge deficits in these patients.

Here, we combine structural, diffusion, and functional MRI (fMRI) and behavioral measures to examine the role of visual–language connections in object knowledge representation in a group of stroke patients. Specifically, we tested whether lesions affecting the connection between the VOTC and language systems affect the local neuro-functional strength of object color representation in the VOTC (Analysis series 1) and whether the connection contributes to supporting object color knowledge behavior (Analysis series 2). We therefore deliberately included stroke patients suffering from lesions outside the VOTC to minimize the confounding effects of local damage to the VOTC on its functionality. Both the patient ($N = 33$) and healthy control groups ($N = 35$) underwent diffusion imaging for white-matter (WM) structural integrity measurement, task-based fMRI experiments for neural representation measurement, and object color knowledge neuropsychological tests for behavioral measurement. Whether and how VOTC-language WM tract integrity affects neural representation in the VOTC-color-knowledge mask and object color behaviors were examined in detail. The analysis scheme is shown in Fig 1.

## Results

Table 1 shows the demographic information of the patient and healthy groups and the lesion information (hemispheric locations and total lesion volume) of the patients (see S1 Table for detailed demographics and lesion coverage of each stroke patient; see S2 Table for the neuropsychological tests of each patient). The lesion distribution of the 33 patients is shown in Fig 1 (upper right panel), which reveals typical lesions in the middle cerebral artery territory, the most frequent stroke location, along with lesions widely distributed in other parts of the brain in some cases (e.g., 8 out of the 33 patients also had lesions in the posterior cerebral artery territory), affecting many WM areas and largely sparing the VOTC (see below for details). A case profile is presented in detail for the purpose of illustrating the whole testing scheme and result pattern (Supporting information, S1 Text and S6 Fig).

### Analysis series 1: Relationship between VOTC-language WM connection integrity and VOTC object color knowledge neural representation

**Defining VOTC and WM masks of interest in healthy controls.** To obtain unbiased results, cortical and WM masks of interest were defined in matched healthy controls, and the patients' data were analyzed in the corresponding masks.

*Object color neural representation in the VOTC—Functional mask defined in healthy controls*: We first identified the object color knowledge representation in the VOTC in healthy controls. The healthy controls participated in an object color judgment fMRI experiment, during which they were presented with grayscale pictures of familiar fruits and vegetables and were asked to judge whether the typical skin color of the fruit/vegetable was red (Fig 2A). We performed representation similarity analysis (RSA) searchlight mapping by correlating the neural representational dissimilarity matrix (RDM, defined as the Pearson distance between the activity patterns of each pair of fruits/vegetables) with the behavior-derived object color representation RDM, while controlling for low-level visual RDM (i.e., gist RDM), shape RDM, and general semantic RDM. As described in the Introduction, our key question focused on object color knowledge effects derived from color-perception processes, which have been the main focus of previous studies [9,10,32]. Following this

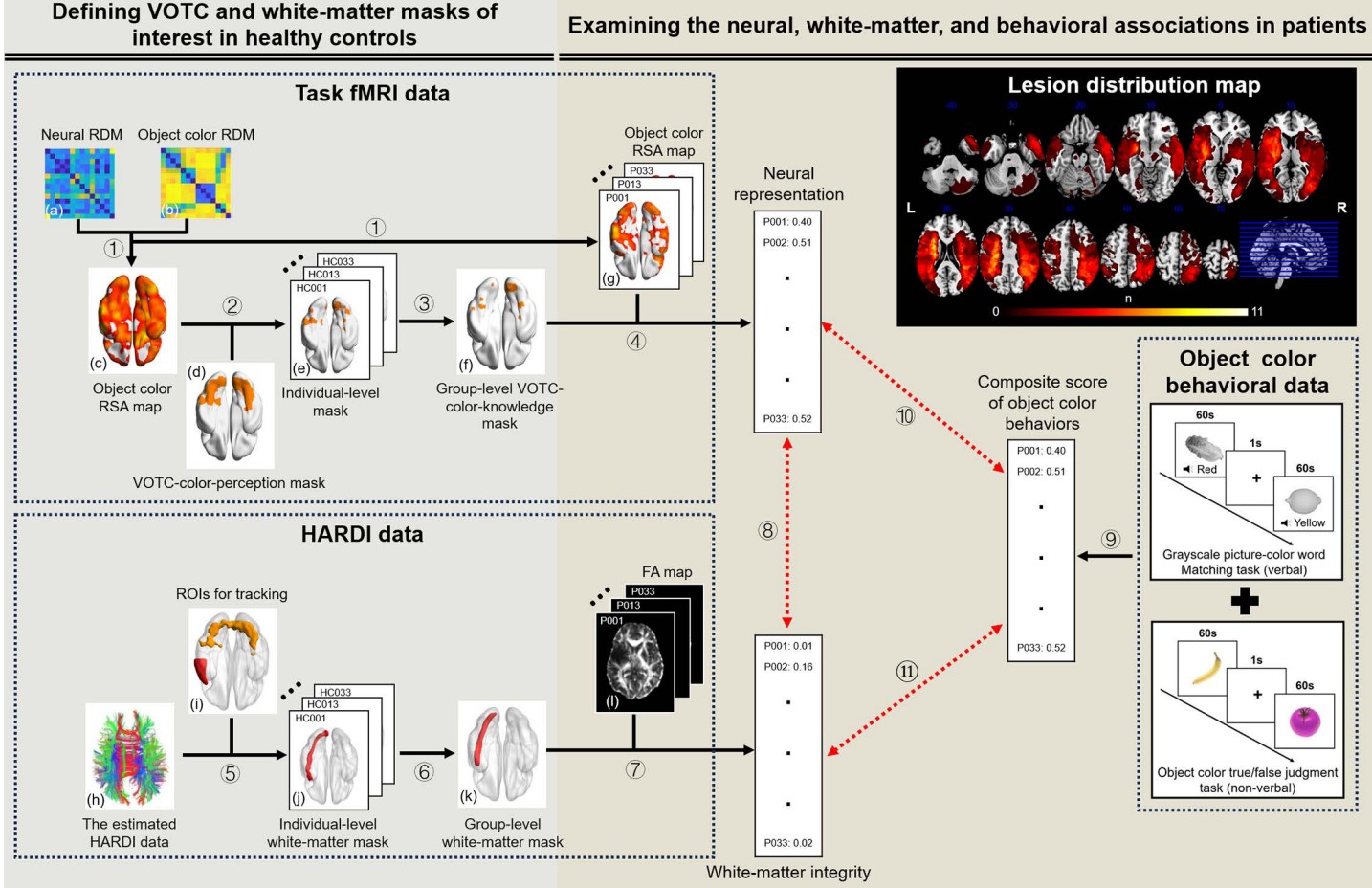

**Fig 1. Flowchart of the imaging and behavioral data analysis and the lesion distribution map of the 33 patients.** The data processing workflow was as follows: **For task fMRI data:** ① The neural RDM (a) was constructed by correlating the activity patterns of each pair of fruits and vegetables (estimated from the task fMRI data) within a sphere (radius = 6 mm) centered on each voxel, via Pearson's correlation distance. The object color RDM of each subject (b) was obtained from pairwise object color similarity ratings. Partial Spearman's rank correlation was then computed between the neural and object color RDMs, controlling for low-level visual control, shape, and general semantic RDMs, to construct the object color RSA maps of healthy controls (c) and patients (g). ② In the functionally defined VOTC-color-perception mask (d), the top 300 selected voxels (i.e., voxels with the highest Fisher-Z transformed $r$ values) in individual participants' object color RSA maps were binarized to construct the individual-level mask (e). ③ The individual masks were overlaid across those of all healthy controls and a group-level threshold of 0.25 was applied to obtain the group-level functional mask (i.e., VOTC-color-knowledge mask, f). ④ The mean Fisher-Z transformed $r$ value of each patient's object color RSA map (g) within the VOTC-color-knowledge mask (f) was calculated to quantify the neural representation of object color knowledge in the VOTC. **For HARDI data:** ⑤ Probabilistic tractography was implemented using the estimated HARDI data (h) between each pair of ROIs (e.g., VOTC-LdlATL, i) in the native space. The resulting tracking map was normalized and standardized using the maximum voxel intensity of each image and then binarized at 0.1 to construct the individual-level mask (j). ⑥ A group-level threshold of 0.5 across all healthy controls within the explicit WM mask was applied to obtain the group-level WM mask (k). ⑦ The mean value of each patient's FA map (l) within the group-level WM mask (k) was calculated to quantify the WM integrity of this fiber bundle. **Correlation analysis:** ⑧ The correlations between the WM integrity (mean FA value) of each WM tract and the neural representation strength of object color knowledge in the VOTC-color-knowledge mask across patients were calculated to identify the tracts relevant to the VOTC knowledge representation. ⑨ Object color behaviors were measured as the composite score of the grayscale picture-color word matching task (verbal) and the object color true/false judgment task (non-verbal). ⑩ ⑪ The correlations were computed between the WM integrity, the neural representation of object color knowledge, and the object color composite score, respectively, to identify the neural correlates associated with object color behavior. The upper-right panel shows the lesion distribution map of the 33 patients. The n value of each voxel denotes the number of patients with a lesion. Brain imaging results were visualized using BrainNet Viewer (version 1.7; https://www.nitrc.org/projects/bnv/; RRID: SCR_009446), MRIcron (version 1.0.20190902; https://www.nitrc.org/projects/mricron; RRID: SCR_002403), or MRIcroGL (version 1.2.20210317; https://www.nitrc.org/projects/mricrogl). *Abbreviations: RDM, representational dissimilarity matrix; VOTC, ventral occipitotemporal cortex; RSA, representation similarity analysis; ROI, region of interest; HARDI, high angular resolution diffusion imaging; L, left; dlATL, dorsolateral anterior temporal lobe; WM, white-matter; FA, fractional anisotropy.*

**Table 1. Demographic data of stroke patients and healthy controls. The data underlying this table are available in S1 Data.**

| | Patient group (*N*=33) | | | Control group (*N*=35) | | | Statistics (*t*) | *P*-value |
|---|---|---|---|---|---|---|---|---|
| | mean | SD | range | mean | SD | range | | |
| Age (year) | 51.55 | 10.02 | 30–65 | 50.23 | 9.94 | 31–65 | 0.54 | 0.59 |
| Gender (M/F) | 25/8 | – | – | 21/14 | – | – | −1.39 | 0.17 |
| Education (year) | 11.18 | 3.26 | 5–16 | 11.71 | 2.79 | 7–16 | −0.72 | 0.47 |
| MMSE (score) | 25.88 | 4.53 | 12–30 | 28.86 | 1.09 | 26–30 | −3.68 | 0.001 |
| Handedness (R/L) | 33/0 | – | – | 35/0 | – | – | – | – |
| Time post stroke (month) | 30.09 | 55.51 | 3–252 | – | – | – | – | – |
| Lesion site (Left/Right/Bilateral) | 18/11/4 | – | – | – | – | – | – | – |
| Lesion volume (mm³) | 44,951 | 55,361 | 852–236,545 | – | – | – | – | – |

Abbreviations: SD, standard deviation; MMSE, Mini-Mental State Examination; M, male; F, female; R, right-handed; L, left-handed.

tradition, the searchlight mapping was performed within a functionally defined, VOTC-color-perception mask (Fig 2B, left panel; defined by contrasting chromatic stimuli to grayscale stimuli in a color perceptual localizer involving 14 healthy controls [22]). Within this VOTC color perception mask, the top 300 voxels (i.e., with the 300 highest *r* values) in the RSA results were chosen for each participant and then overlaid across all the healthy controls (*N*=33). This yielded a probability map (Fig 2B, middle panel), where a warmer color represented voxels that were more consistent across healthy controls in terms of representing object color knowledge. Voxels that had a probability greater than 0.25 on the map, distributed in the bilateral fusiform and lingual cortex, were retained as a functional object-color-knowledge mask for the patient analyses below (hereafter referred to as the VOTC-color-knowledge mask; see Fig 2B, right panel). This functional mask is largely in line with those reported in previous studies about object color knowledge, especially those using picture stimuli ([6,7,11,33–35]; see S7 Table). Given that the selection of a functional mask is typically arbitrary, we carried out validation analyses using functional masks with different combinations of individual-level thresholds (top 200−500 voxels, staircase 50 voxels) and group-level probability thresholds (range: 0.15–0.35, staircase 0.05) and obtained largely consistent results (S4 Table). The group-level whole-brain analyses in the healthy controls also revealed object color knowledge effects in the VOTC, with the peak located in the right lingual gyrus (voxel-level *p*<0.001, S1 Fig).

***WM connections between the VOTC and language regions—Tracking results in healthy controls:*** To map the WM connections between the left language regions and the VOTC, we performed probabilistic tractography on healthy controls (*N*=35) using high angular resolution diffusion imaging (HARDI) data, between the six language seed regions (by contrasting intact sentences to nonword lists in 220 healthy controls; see Fedorenko and colleagues [36] for details) and the VOTC-color-perception mask shown in Fig 2B (left panel): VOTC-left dorsolateral anterior temporal lobe (LdlATL), VOTC-left posterior middle temporal gyrus (LpMTG), VOTC-left angular gyrus (LAG), VOTC-left inferior frontal gyrus, orbital part (LIFGorb), VOTC-LIFG, and VOTC-left middle frontal gyrus (LMFG). As shown in Fig 2C (left panel) and S3 Fig (top panel), the fiber tracking was successful for all six connections, with an individual-level threshold of 0.1 for the probabilistic fiber tracking and a group-level threshold of 0.5 for the fiber tracking success rate (further explicitly masked by the SPM12 WM map with probability>0.4), aligning well with the major WM bundles (S3 Table). Note that while we used a bilateral VOTC color mask, under current thresholds for obtaining the WM tracts, no tracts connecting the right VOTC mask and the left language regions were included. Those tracts connecting the right VOTC mask and the left language regions through the corpus callosum could be observed when the individual level threshold was loosened to 0.001 (see S2 Fig for a full WM connection probabilistic map between the bilateral VOTC and the dlATL, for example). These reconstructed WM tracts in the healthy control group served as the main tracts of interest for the analyses of the patients' data below. To confirm the robustness of the results, we conducted validation analyses using WM tract masks with various

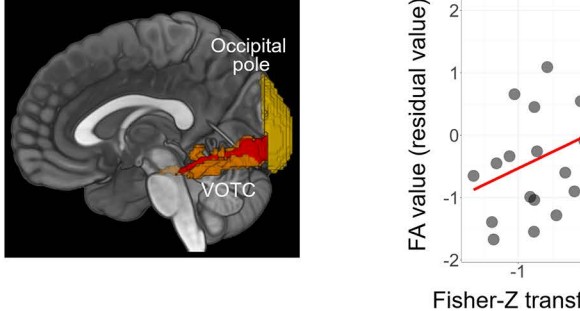

**A** In-scanner task: Is the typical color of its skin red?

**B**

Unique object color probability map

> 0.25

VOTC-color-perception mask

Proportion of 33 healthy controls

VOTC-color-knowledge mask

**C** Correlation between the VOTC-LdlATL tract integrity and the VOTC neural representation

rho = 0.56
FDR q = 0.005

**D** Validation 1: controlling for seed GM damage &TLV

rho = 0.53
p = 0.003

**E** Validation 2: controlling for the visual perception pathway integrity & TLV

rho = 0.53
p = 0.002

**Fig 2. Analysis series 1: Relationship between VOTC-language white-matter connection integrity and VOTC object color knowledge neural representation. (A)** Design of the object color judgment task fMRI experiment. Participants viewed the grayscale pictures of fruits and vegetables (1 s)

and pressed the "yes" button when the skin of the items was red, and the "no" button when it was not. **(B)** Defining the VOTC-color-knowledge mask in healthy controls. The left panel displays the VOTC-color-perception mask, functionally defined by contrasting chromatic stimuli to grayscale stimuli in a color perceptual localizer in 14 healthy subjects [22]. The middle panel shows the unique object color probability map, constructed by the overlap of individual ROIs (i.e., the top 300 selective voxels with the highest Fisher-Z transformed $r$ values of the color RSA results within the VOTC-color-perception mask) across 33 healthy controls. The color bar indicates the proportion of healthy controls that had the individual ROI at each voxel. Voxels with a probability greater than 0.25 (outlined by the black lines) were retained as the VOTC-color-knowledge mask, as depicted in the right panel. **(C)** The left panel illustrates the reconstructed VOTC-LdlATL tract (red) and the seed regions being connected (VOTC, orange; LdlATL, green). The right panel shows the raw scatter plot between the mean FA values of the VOTC-LdlATL tract and the VOTC neural representation (Fisher-Z transformed $r$ values) and the partial Spearman's rho value (controlling for TLV). Note that negative values for the RSA results might be difficult to interpret; when these values were set to zero, the effect remained significant (partial rho = 0.54, $p < 0.01$). **(D)** Validation of the VOTC-LdlATL tract after controlling for GM damage at both ends and TLV. **(E)** Validation of the VOTC-LdlATL tract after controlling for the VOTC-occipital pole tract integrity and TLV. The reconstructed VOTC-occipital pole tract (red) and the seed regions (VOTC, orange; occipital pole, yellow) being connected are shown in the left panel. The data underlying this figure are available in S1 Data. Brain imaging results were visualized using BrainNet Viewer (version 1.7; https://www.nitrc.org/projects/bnv/; RRID: SCR_009446), or MRIcroGL (version 1.2.20210317; https://www.nitrc.org/projects/mricrogl). *Abbreviations: VOTC, ventral occipitotemporal cortex; L, left; dlATL, dorsolateral anterior temporal lobe; ROI, region of interest; TLV, total lesion volume; GM, gray-matter; FA, fractional anisotropy.*

combinations of individual-level probability thresholds (0.05–0.2, staircase 0.05) and group-level probability thresholds (0.35–0.6, staircase 0.05), and with or without the explicit WM mask. The patterns of the results were largely consistent with the main findings (S5 Table).

**Reduced VOTC-language WM connection integrity associated with deteriorated VOTC object color neural representation in patients.** The key question here was whether the integrity of the WM tracts connecting the left language system and VOTC was necessary for the functional neural representation of object color knowledge in the VOTC, beyond the general lesion severity of the stroke patients. Given that stroke-related damage may extend beyond visible lesions [37–40], we focused on the fractional anisotropy (FA), which, by measuring the directional constraint of water diffusion, is commonly considered to be a sensitive measure of the structural integrity of WM fibers.

First, in parallel with the healthy control group, the patients also participated in the task fMRI scanning, i.e., to judge whether the typical skin color of the fruit/vegetable in the grayscale picture was red. Note that all patients understood "red" as they correctly matched the color word "red" to the red patch in a color patch matching task performed outside the scanner (see S4 Fig). This task and stimulus selection was tailored to ensure that the patients could perform with ease, so that meaningful VOTC neural activity could be measured (see Materials and methods for details). In-scanner behavioral response profiles were comparable at the ceiling for patients and healthy controls (accuracy of controls: 0.92 ± 0.06 (standard deviation, SD); accuracy of patients: 0.90 ± 0.09 (SD); $t$ (64) = 1.13, $p = 0.26$).

Then, partial Spearman's rank correlation analyses were conducted between the mean FA values of each language-VOTC tract (HARDI results) and the color knowledge representation strength in the VOTC-color-knowledge mask (defined in healthy controls above; object color RSA results controlling for low-level visual, shape, and general semantic features), with total lesion volume as a covariate. We noted an outlier in the correlation analysis: one patient (ID: 027) had Fisher-Z transformed $r$ values of the RSA results beyond +3.3 SD of the stroke patient group. We performed analyses using both the entire patient group and a subset excluding the outlier patient, and the results were highly consistent. We report the results of the whole group analysis below for simplicity.

As shown in Figs 2C and S3, the integrity of the tracts connecting the LdlATL and VOTC, as well as the left pMTG and VOTC, was significantly positively correlated with the VOTC object color knowledge representation (i.e., Fisher-Z transformed $r$ values of the RSA results), beyond the effect of total lesion volume (VOTC-LdlATL: partial rho = 0.56, FDR $q < 0.01$; VOTC-LpMTG: partial rho = 0.46, FDR $q < 0.05$). Given that the VOTC-LdlATL and VOTC-LpMTG connections overlapped, we carried out a specificity test by excluding overlapping voxels and considering the mean FAs of the tract-specific voxels. The effects of the VOTC-LdlATL connection and the VOTC-LpMTG connection remained significant (VOTC-LdlATL: partial rho = 0.52, $p = 0.003$; VOTC-LpMTG: partial rho = 0.38, $p = 0.03$), confirming that their contribution to the VOTC functional representation of object color knowledge is specific relative to each other. However, while

the effects of the VOTC-LdlATL connection were significant even when partialling out the effects of the VOTC-LpMTG connection (partial rho = 0.36, $p < 0.05$), the reverse is not true (partial rho = 0.03, $p = 0.89$), indicating that the effects of the VOTC-LdlATL connection were most robust. Considering that negative correlations for RSA are difficult to interpret, in an additional analysis, we assigned zero to the negative Fisher-Z transformed $r$ values of the RSA results, and the VOTC-LdlATL connection result remained significant (partial rho = 0.54, $p < 0.01$). When the VOTC in the two hemispheres were analyzed separately, both neural representations were significantly correlated with the integrity of the VOTC-LdlATL tract (left VOTC: partial rho = 0.38, $p = 0.03$; right VOTC: partial rho = 0.51, $p = 0.003$).

Taken together, the integrity of the VOTC-LdlATL WM connection has robust effects on how well the VOTC represents object color knowledge.

**Specificity of the effects of the VOTC-LdlATL WM connection on VOTC neural representation in patients.** Is the effect of the VOTC-LdlATL connection integrity on the VOTC knowledge representation strength specific to the WM connection to the LdlATL, or could it be explained by alternative variables? First, in all the analyses above, we controlled for the effect of total lesion volume; thus, the overall size of the stroke could not explain the effects of the VOTC-LdlATL connection integrity. We carried out the following series of analyses to examine other potential alternative neuroanatomical and/or cognitive variables that may contribute to the observed effects of interest.

*Is the WM integrity effect on VOTC functional representation driven by damage to the gray-matter (GM) regions being connected?* We included the lesion percentage in the GM regions at both ends of each connection of interest as a covariate in addition to total lesion volume, and the effects of the tract connecting the VOTC and LdlATL were still significant (Fig 2D, partial rho = 0.53, $p < 0.01$).

*Is the WM integrity effect on VOTC functional representation driven by deficits in the early visual perception pathway?* The object color knowledge representation strength of the VOTC functional cluster may be contingent upon the integrity of the visual processing stream from the early visual cortex to the higher visual cortex (VOTC). While our patients did not have any lesioned voxels in the VOTC-color-knowledge mask (see Materials and methods), it is still possible that the integrity was affected by lesions in the WM connections with the early visual cortex. Therefore, we reconstructed the fiber tract between the VOTC and the occipital pole (OP) in healthy controls (Fig 2E, left panel) and extracted the mean FA values for each stroke patient. Partial Spearman's rank correlation analysis revealed that the VOTC-OP tract did not significantly correlate with object color knowledge representation in the VOTC (partial rho = 0.23, $p = 0.21$). We further added the mean FA values of this tract as a covariate, along with total lesion volume, for the correlation between the VOTC-LdlATL and the VOTC neural representation. The effect of the VOTC-LdlATL was still significant (Fig 2E, right panel; partial rho = 0.53, $p < 0.01$).

*Is the WM integrity effect on VOTC functional representation contaminated by patient etiology?* A few special patient cases warrant scrutiny. As explained in the Materials and methods section, one patient (ID: 018) with a premorbid distance vision problem (not improved with correction) completed the auditory version of task-state fMRI scanning, where the stimuli were auditory words (names of the fruits/vegetables) with otherwise the same presentation parameters as the visual version. To eliminate the influence of modality, we excluded this patient and the effect among the remaining patients was still significant (partial rho = 0.56, $p < 0.001$). Additionally, while first-time stroke was among our patient inclusion criteria, two patients' scans (ID: 003, 007) revealed traces of potential old lesions. We performed analyses excluding these two patients, and the effect of the integrity of the VOTC-LdlATL connection remained significant (partial rho = 0.55, $p < 0.01$). The same was true when regressing out total lesion volume and post-onset time (number of months) across patients (partial rho = 0.52, $p < 0.01$).

In summary, the integrity of the WM connection between the VOTC and the LdlATL (in the language mask) robustly influences the neural representation of object color in the VOTC, the effects of which are not explained by broader effects of lesions, the early visual perception pathway, specific stroke etiologies, or patient post-onset duration.

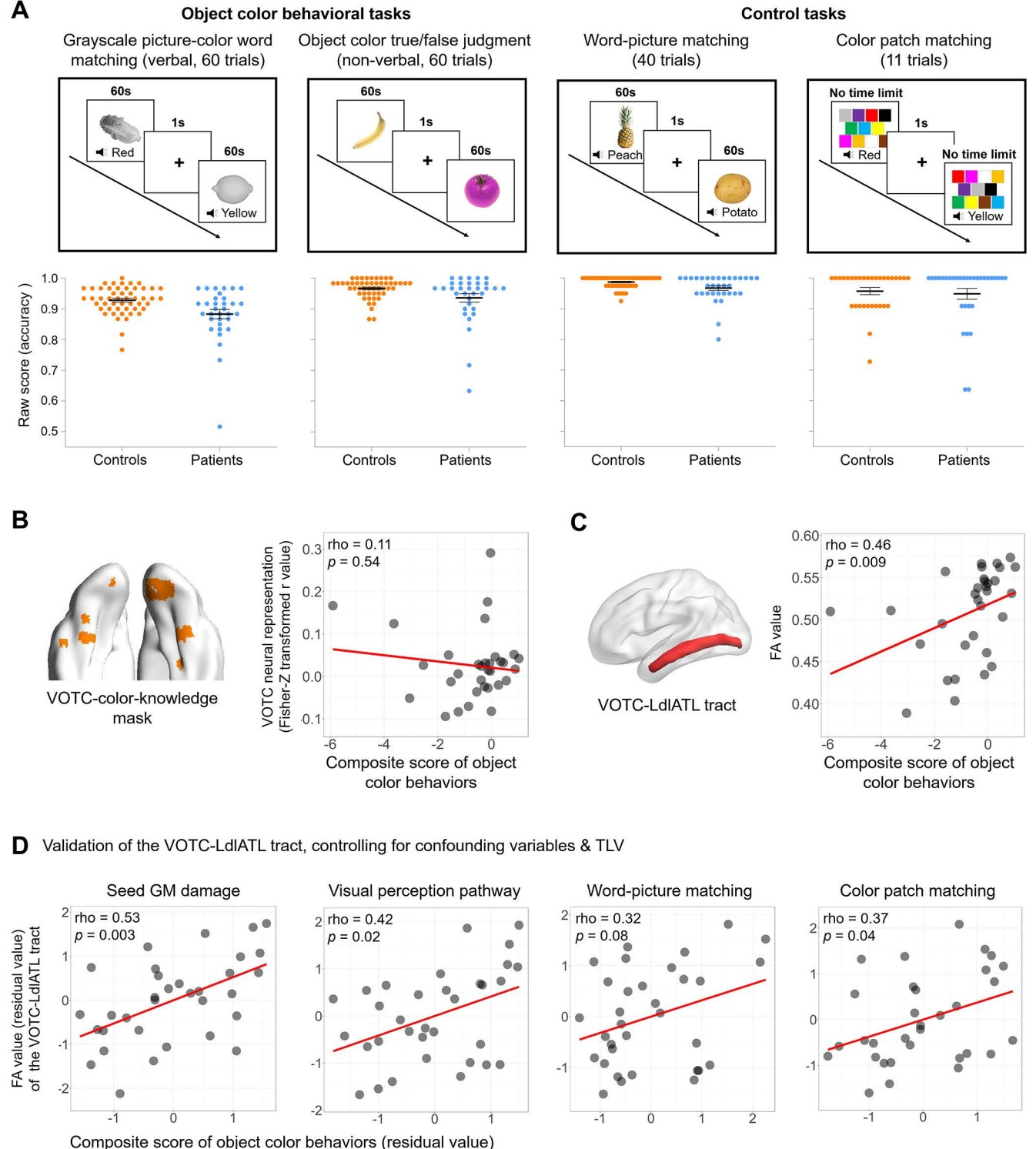

**Fig 3. Analysis series 2: Relationship between VOTC-language white-matter connection integrity and object color knowledge behavior in patients. (A)** Design and raw accuracy in the out-scanner neuropsychological tests, including object color behavioral tasks (grayscale picture-color word matching (verbal), object color true/false judgment (non-verbal)) and control tasks (word-picture matching, color patch matching). Dot plots show raw accuracies of each participant (bars denote mean values ±1 standard error). **(B, C)** The relationships between two neural measures (the neural

representation of object color knowledge in the VOTC-color-knowledge mask, and the VOTC-LdlATL tract integrity) and object color behavior (the composite score across the verbal and non-verbal object color tasks). The scatter plots are shown for visualization purposes, as we removed an outlier patient (ID 006, composite score = −10.45, 4.3 SD below the patient average; mean FA value = 0.53) from the plots, the presence of which greatly increased the range of the x-axis and distorted the scatter plot. Importantly, the partial rho and p-values shown in the figures were computed on the basis of data from all the patients, after controlling for TLV. The correlations after excluding this outlier patient were similar for the neural representation (partial rho = 0.19, p = 0.30) and for the VOTC-LdlATL tract (partial rho = 0.48, p = 0.007). **(D)** Validation analyses of the VOTC-LdlATL tract, controlling for confounding variables and TLV. The data underlying this figure are available in S1 Data. Brain imaging results were visualized using BrainNet Viewer (version 1.7; https://www.nitrc.org/projects/bnv/; RRID: SCR_009446), or MRIcroGL (version 1.2.20210317; https://www.nitrc.org/projects/mricrogl). *Abbreviations: VOTC, ventral occipitotemporal cortex; L, left; dlATL, dorsolateral anterior temporal lobe; TLV, total lesion volume; FA, fractional anisotropy; GM, gray-matter.*

## Analysis series 2: Relationship between VOTC-language WM connection integrity and object color knowledge behavior

**Reduced VOTC-LdlATL WM connection integrity associated with object color behavior impairments in patients.** The above analyses revealed that the integrity of the WM connection between the LdlATL and the VOTC in patients affected the functionality of the local VOTC neural representation of object color knowledge. A follow-up key question is how they relate to behavior, which is examined below.

*Patient neuropsychological test performance*: For object color knowledge, patients performed two object color knowledge tasks that varied by input modality-verbal and non-verbal-with otherwise identical item and task structures (Fig 3A, top left): the grayscale picture-color word matching task (verbal) and the object color true/false judgment task (non-verbal). To consider whether the patients' deficits in performance on the two core object color knowledge tasks were due to impairments in peripheral processes—namely, the ability to recognize objects in both tasks and to recognize color names in the verbal object color task—we evaluated their performances on a word-picture matching task and a color patch matching task (Fig 3A, top right, control tasks).

Fig 3A (bottom panel) shows the raw accuracies of these neuropsychological tests for each of the healthy controls and stroke patients (see S2 Table for the detailed behavioral scores of each patient in each task). Each patient's standardized scores of these tests were computed as the point estimates of effect sizes based on the single-case Bayesian test for deficit [41], which controls for the variation in demographic factors (i.e., age, gender, and years of education) among the two groups. We then computed the correlations of these behavioral scores with the neural color representations in the VOTC and the VOTC-LdlATL tract integrity, with total lesion volume as a covariate. A composite score across the two object color tasks was considered in the main analyses, with the results of each individual task shown in S4–S6 Tables (no significant differences were found between the verbal and non-verbal object color tasks across all analyses). Note that while in the patient population, the extreme values were not necessarily outliers, the resulting patterns reported below were robust even when the outliers defined in the conventional sense were excluded (ID 027, VOTC *r* value +3.3 SD; ID 006, behavioral composite score −4.3 SD), and are not shown in detail for the sake of simplicity.

*VOTC and object color knowledge performance*: For the VOTC-color-knowledge mask, the strength of the object color neural representation was not significantly correlated with the composite object color behavioral scores (Fig 3B; partial rho = 0.11, p = 0.54).

*VOTC-LdlATL tract integrity and object color knowledge*: The VOTC-LdlATL WM tract integrity was significantly correlated with composite object color knowledge scores across patients (Fig 3C; partial rho = 0.46, p < 0.01). The significant effect of this connection in explaining behavior was beyond the potential effect of the VOTC local functionality; after controlling for the VOTC representation strength, the effects remained significant (partial rho = 0.48, p < 0.01). The correlations between object color behavior and the integrity of tracts connecting the VOTC to other language seeds were not significant (S3 Fig, bottom panel; partial rhos < 0.28, ps > 0.12). The VOTC-LdlATL tract integrity was significantly correlated with the verbal object color behavior (rho = 0.41, p = 0.02) and marginally correlated with non-verbal object color behavior (partial rho = 0.30, p = 0.099), with no significant difference between these two tasks (Hotelling's *t* values = 0.61, p = 0.55; S6 Table).

Finally, given that the healthy control participants were selected to be demographically matched with the patients, they spanned a wide range of ages (from 31 to 65 years) and years of education (from 7 to 16), and showed decent variability in the neuropsychological test performance. Therefore, we carried out exploratory Spearman's rank correlation analyses between the mean FA value of the tracts connecting the VOTC and left language regions and the object color composite score across the healthy controls ($N = 34$). Only the FA value of the VOTC-LdlATL tract was significantly positively correlated with the object color composite score (VOTC-LdlATL: partial rho $= 0.42$, $p = 0.01$; other tracts: partial rhos $< 0.31$, $p$s $> 0.07$). These results converge with the patient findings and highlight the role of the VOTC-LdlATL tract in supporting object color knowledge behaviors.

**Testing the specificity of the VOTC-LdlATL tract effect in object color knowledge behavior.** *Specificity of the tract: Is the WM integrity effect on object color knowledge behavior driven by broader neural correlates?* In this set of analyses, we found that the relationship between the VOTC-LdlATL tract integrity and object color knowledge behavior was not explained by the broad effects of the lesions. We controlled for the total lesion volume in all the analyses above; thus, the overall size of the stroke could not account for the effect of the VOTC-LdlATL connection integrity on object color behavior. To consider the GM nodes connecting this tract of interest, we added the lesion percentage on the GM regions at both ends of this connection as covariates along with the total lesion volume. The effects of the tract connecting the VOTC and LdlATL remained unchanged. The FA value of the tract positively was correlated with the composite score ([Fig 3D](), the leftmost subplot; partial rho $= 0.53$, $p < 0.01$).

We then considered whether the correlation was affected by lesions in the early visual perception stage. When the FA value of the VOTC-OP tract was included as a covariate, the relationship between the VOTC-LdlATL tract integrity and object color knowledge behavior remained significant ([Fig 3D](), the second subplot; partial rho $= 0.42$, $p < 0.05$).

*Specificity of cognitive processes: Is the WM integrity specifically associated with object color knowledge relative to other cognitive tasks?* Here, we examine the extent to which the VOTC-LdlATL integrity reduction was associated with object color knowledge specifically. The two most relevant tasks were considered—object recognition and color recognition (word-picture matching and color patch matching). The VOTC-LdlATL tract was positively correlated with object recognition and color patch recognition performance (word-picture matching: partial rho $= 0.36$, $p < 0.05$; color patch matching: partial rho $= 0.38$, $p < 0.05$), suggesting its association with processing objects and color in general. Critically, when controlling for object recognition and color recognition performance, in addition to total lesion volume, the VOTC-LdlATL tract integrity was still significantly or marginally significantly associated with the object color composite score ([Fig 3D](), the third and rightmost subplots; word-picture matching: partial rho $= 0.32$, $p = 0.08$; color patch matching: partial rho $= 0.37$, $p < 0.05$). That is, the contribution of the tract to object color knowledge representation is above and beyond its contribution to recognizing objects and color patches.

*Consideration of patients' post-onset time and patients with specific stroke etiologies:* First, the results were not affected by post-onset time differences, as when we added the post-onset time as an additional covariate besides the total lesion volume, the association between the VOTC-LdlATL tract integrity and object color behavior remained significant (partial rho $= 0.44$, $p < 0.05$). The results also remained robust when we excluded the two cases who had suspected old lesions (ID: 003, 007), similar to the section above (partial rho $= 0.43$, $p < 0.05$).

In summary, the integrity of the VOTC-LdlATL connection, not the local VOTC functionality, robustly influences object color behavior, the effects of which are not explained by broader effects of lesions, related cognitive processes, patient post-onset duration, or specific stroke etiologies.

### In-depth analyses of the VOTC-LdlATL connection

Having established the important role of VOTC-LdlATL tract integrity in both VOTC neural representation and behavioral profiles of object color knowledge, in this section we look closer at this tract, examining whether the effects were: (1) mainly driven by any specific subsection along the tract; (2) lateralized on the left hemisphere; (3) specific to the dorsolateral subregions of the ATL; (4) present in patients with only left hemisphere lesions.

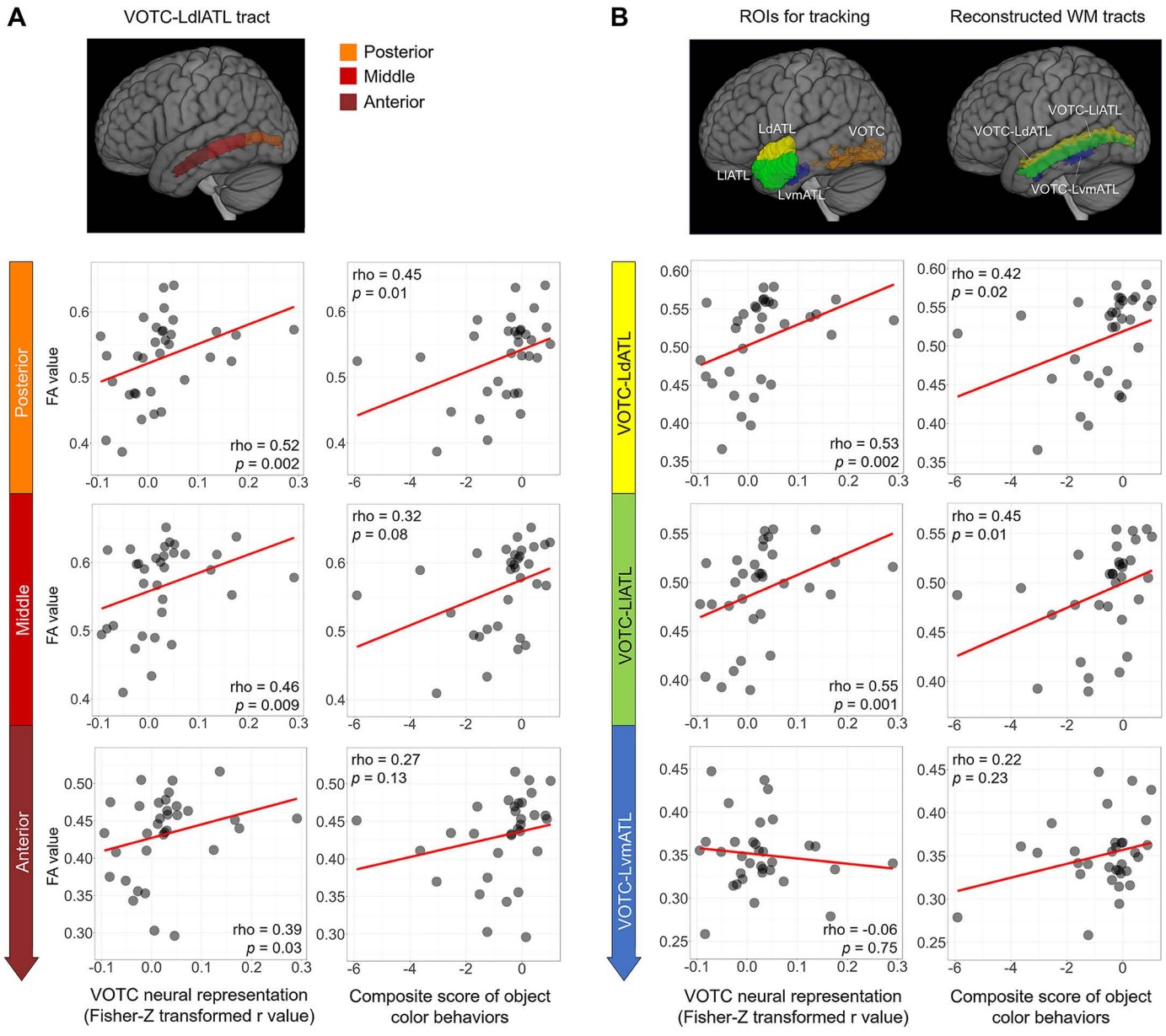

**Fig 4. In-depth analyses of the VOTC-LdlATL connection. (A)** Effects of subsections along the VOTC-LdlATL tract. The tract was divided into the three equal subsections along the y-axis (top panel): posterior, middle, and anterior. The bottom panels show the scatter plots of the FA values of these subsections with the VOTC object color neural representation and the composite score of object color behaviors, respectively. **(B)** Specificity of the dorsolateral subregion of the ATL. The ATL subregions (dorsal, lateral, and ventral-medial) were obtained from Hung and colleagues [43], who parcellated the ATL on the basis of coactivation clustering; the two dorsal subregions in that study were combined here into a single dorsal subregion. The top panel displays the seed regions for probabilistic tractography and the corresponding reconstructed white-matter tracts. The bottom panels show the scatter plots of FA values of the reconstructed white-matter tracts with the VOTC object color neural representation and the composite score of object color behaviors, respectively. The partial rho and *p*-values shown in the figures were computed with all the patients, after controlling for total lesion volume. Similar to Fig 3, the behavioral outlier patient was removed from the scatter plots with object color behavior. The data underlying this figure are available in S1 Data. Brain imaging results were visualized using MRIcroGL (version 1.2.20210317; https://www.nitrc.org/projects/mricrogl). *Abbreviations: VOTC, ventral occipitotemporal cortex; L, left; dlATL, dorsolateral anterior temporal lobe; FA, fractional anisotropy.*

**Effects by subsections along the VOTC-LdlATL tract.** In this analysis, we divided the VOTC-LdlATL tract into three equally sized subsections on the y-axis: posterior, middle, and anterior. As shown in Fig 4A, the associations with both VOTC neural representation and the composite behavioral score of object color knowledge were significant or showed a similar trend across all three subsections, indicating that this tract functions as an integral neural substrate to support color knowledge representation.

**Lateralization effects of the VOTC-LdlATL connection on VOTC neural representations and object color behaviors.** In this analysis, we examined whether the right homologous VOTC-dlATL connection has similar effects as the connection in the left hemisphere. Correlation analyses revealed that the VOTC-RdlATL connection did not significantly correlate with the VOTC neural representation of object color knowledge (partial rho = −0.01, *p* = 0.94). The VOTC-RdlATL connection did not correlate with object color behavior (partial rho = −0.08, *p* = 0.68). These results support the left lateralization of the VOTC-LdlATL connection in its role in supporting object color knowledge.

**Specificity of the dorsolateral subregion of the ATL.** The observed effective tract connects the VOTC with the dorsolateral aspect of the ATL, predefined in the language mask. Rich evidence has shown that different subdivisions of the ATL serve different role in aspects of semantic representation [5,21,42,43]. Therefore, we also performed WM tracking between the VOTC and different ATL subregions, parcellated either by the anatomical profile (i.e., Harvard-Oxford Atlas (probability > 0.2, see Materials and methods for details)) or functional activation connectivity profile [43], in the healthy group. The results were highly similar when these two types of parceling were used, indicating that the connections between the VOTC and the dorsal and lateral (not ventral) ATL subregions indeed drove the effects we observed (see Figs 4B and S5).

Finally, when patients with only left hemisphere lesions (*N* = 18) were considered, the key analyses revealed a pattern comparable to that of the whole group, although not reaching statistical significance for the behavioral analyses due to reduced *N* (partial rho value between VOTC-LdlATL connection integrity and VOTC neural RSA strength = 0.53, *p* = 0.03; between VOTC-LdlATL connection integrity and object color knowledge behavior score = 0.43, *p* = 0.09).

## Discussion

Combining HARDI data, task fMRI data on object color processing, and object color behavioral data from 33 stroke patients, we made two main discoveries. First, across patients, the structural integrity of the VOTC-LdlATL connection exerts a robust effect on the neural representation strength of object color knowledge in the VOTC. Second, VOTC-LdlATL tract integrity accounts for object color knowledge behavioral task performance beyond the potential effects of VOTC representations. This effect is above and beyond its contribution to related cognitive processes (i.e., object recognition and color patch recognition). These two lines of results remained robust when we controlled for a wide range of potential confounding variables, including the broader effects of lesions, patient post-onset duration, and patients with specific stroke etiologies, and were not attributable to potential associative impairment in the early visual perception pathway.

Our main results nicely accommodate the previous imaging and patient lesion evidence and offer critical clarifications. As described in the Introduction, object color representation has been observed in healthy individuals in both the VOTC, presumably derived from sensory experience, and the dorsal ATL, presumably derived from non-sensory, language experience (see review in Bi [24]). Whether the commonly assumed sensory-derived representations function independently and sufficiently to support object knowledge remains unresolved. Object color knowledge deficits have been reported in stroke cases with ventral occipitotemporal lesions (spanning anteriorly) and in semantic dementia patients with ATL lesions, indicating that neither the posterior representation nor the anterior representations might be sufficient to support intact object color knowledge behavior [12,14,27,28,44]. The results of a case study showing impaired color name comprehension following a stroke that functionally disconnected color-biased visual regions from the dorsal and lateral parts of the left ATL, are suggestive of the potential role of the VOTC-LdlATL connection [45]. Inspection of lesion locations across these stroke cases and semantic dementia patients revealed that their lesions may affect the VOTC-LdlATL WM tract to

varying degrees. However, conclusions about implicated neural structures are difficult to draw, given the intrinsic difficulty of case studies.

By showing that, indeed, this connection disruption leads to VOTC functional representation reduction and has unique explanatory power for object color knowledge behavioral deterioration, our findings well explain the batch of patient findings and answer questions posed by the dual knowledge systems [24]. Note that while the VOTC functional integrity was significantly modulated by its connection with the LdlATL, its correlation with object color behavior was not significant. This null effect should be interpreted with caution. As we deliberately selected stroke patients whose lesions largely spared the VOTC, this patient cohort may not be optimal for revealing the effects of the VOTC. Another possibility is that behavior is the readout of many types of representations, and the performance for judging object color knowledge tested here may (also) pull from other types of representations stored elsewhere, i.e., the integration of such VOTC representations and those in the LdlATL. The key point is that, in explaining behavior, the effects of the VOTC-LdlATL connection are beyond the potential effects of VOTC functionality.

Can our findings regarding the VOTC-LdlATL connection simply be because the fMRI task involves mapping to color words, i.e., mapping object color knowledge in the VOTC to color words represented in the ATL? We reason that this is unlikely. The fMRI task was designed to require minimal verbal retrieval—only the color name "red". The instruction was to judge whether the object depicted in the grayscale image was usually red. All patients had no difficulty recognizing the color red, as shown in the color word matching with the color patch task (see S4 Fig). By the null hypothesis that VOTC representation is sufficient, the patients could accomplish the task by activating the object color knowledge representation in the VOTC, comparing it with the red color representation there, and making a correct response. Furthermore, in the object color behavioral assessments, we also included a non-verbal task (true/false color judgment), where mapping to words is not intrinsic to the task. Yet we observed that the disruption of the VOTC-LdlATL connection not only affected VOTC functionality but also contributed to color knowledge behavior independently.

What is the functionality of the observed VOTC-LdlATL connection? The connection observed here was obtained by WM tracking between color-perception clusters in the VOTC and the subregion of the ATL parcellated from the language-activation mask [36] in healthy controls. This particular tracking to the language system was motivated by the findings about the language-derived representations here [22]. Our findings on the connection between the VOTC and the LdlATL, and not the other language parcels in the frontal and posterior temporal regions, converge with previous findings on identifying disembodied and language-experience-related knowledge representations in the dorsal ATL [21–23], highlighting the central role of this particular neural structure, more so than the other language regions, in interfacing with the sensory-derived representations to support knowledge. Rich evidence has shown that different subdivisions of the ATL serve different aspects of semantic representation, with the ventral-medial region being more related to the multimodal binding of object knowledge and the dorsolateral aspect being more related to language-derived abstract representations ([21,22,43]; see Bi [24], Lambon Ralph and colleagues [5] for review). Indeed, our ATL-subregion analyses confirmed that the connection between the VOTC and the dorsolateral ATL is indeed effective, although the negative effects of the ventral ATL should not be overinterpreted due to suboptimal lesion coverage and imaging signals (Figs 4B and S5). To better situate the target tract in the larger WM pathways, we overlaid the target connection with the major WM tracts in the Johns Hopkins University (JHU) WM template and found that it partly comprises the inferior longitudinal fasciculus (ILF; 50.6% overlapping voxels), the inferior fronto-occipital fasciculus (IFOF; 24.1% overlapping voxels), and, at minimum, the forceps major (2.9%). These results are in accordance with results of previous studies showing that these large WM bundles are involved in supporting object knowledge: the IFOF has been consistently shown to be involved in higher-order object semantic processing across multiple modalities [46–48], and the ILF tends to be implicated in non-verbal object processing tasks [49,50], although its role in more abstract semantic relationships or mapping with object words is less clear [51–53].

It has been shown that such long-range WM connections are one candidate that supports high-dimensional semantic space by incorporating cortical representations that process different specific dimensions of information [54]. Consistent with this broad semantic function of these WM connections, we also observed that the integrity of the VOTC-LdlATL connection contributes significantly to object recognition performance. Importantly, the VOTC-LdlATL connection integrity explained object color knowledge beyond such object recognition effects. Considering the previous findings about different types of object color knowledge representations (non-sensory language derived versus color sensory derived; see review in Bi [24]), one likely mechanism of this connection is to form an integral object color knowledge representation by incorporating different types of color representations (sensory-derived in the VOTC and language-derived in the LATL). It offers a solution to the question posed by the dual knowledge representation framework [24,55,56], i.e., how two different forms of representations are integrated. This observation extends the previous notion of representing concepts by connecting different attributes [3], to representing object knowledge by connecting different representation forms of the same attribute. Another, non-mutually exclusive possibility, is that language representations play an online modulatory role in categorizing object properties, especially along perceptual dimensions such as color, where other differences need to be abstracted away [57,58]. The results also, more broadly, shed light on a classical debate of the effects of language on cognition [e.g., 19,59,60]. By providing positive evidence for the effects of the vision-language connection on object (non-verbal) knowledge representations, the results speak against a strong modular view of the two systems [e.g., 60]. These findings, by highlighting the role of perception-language system connections in knowledge loss, motivate rehabilitation strategies that target such components, such as cognitive trainings that involve vision-language-alignment and/or brain stimulation targeting the underlying connectional functionality.

As in all lesion studies, the power of our analyses is constrained by the lesion coverage of the patient group. Here, we specifically avoided lesions affecting the VOTC in the interest of examining the contribution of non-posterior connections. The resulting group tended to exhibit a typical lesion distribution of middle cerebral artery strokes, with comparable impairment in connections of interest, and making inferences about other connections should be refrained.

To conclude, addressing the question of object knowledge representation and taking object color as a test case, the neural representation in the sensory-derived VOTC is modulated by the disruption of its connection with the LdlATL in the language network, which also explains object color knowledge behavior better than the VOTC neural representation does. These results underscore the critical role of the VOTC-LdlATL connection in representing object knowledge, implicating its specific involvement in bridging the language-derived representation and the sensory-derived knowledge representation, which in turn modulates VOTC functionality. These findings highlight the intricate interaction between vision and language systems in the human brain and the importance of studying brain connections in supporting knowledge representation [54].

## Materials and methods

### Participants

Thirty-three stroke patients and 35 healthy controls participated in the fMRI and behavioral experiments (see Table 1 for the demographic data of the two groups). All participants were native Chinese speakers who received payments and provided written informed consent. The study has been conducted according to the principles expressed in the Declaration of Helsinki, and was approved by the Ethics Committee of First Hospital of Shanxi Medical University (Approval No. 2021-K035).

Thirty-three stroke patients (25 males), without major lesions affecting the VOTC, were recruited from the First Hospital of Shanxi Medical University. The inclusion criteria were as follows: age 20–65 years; right-handed before stroke onset [61]; normal vision or corrected vision; at least 3 months after stroke; first symptomatic stroke (ischemic or intraparenchymal hemorrhagic); no contraindications for MRI; lesion location involving the cortex and/or subcortical WM and not the

VOTC; no other neurological or psychiatric diseases; ability to perform simple cognitive tasks and understand instructions; and intact color vision on the color vision examination plates test [62, similar to the Ishihara test]. The Chinese version of the Mini-Mental State Examination [MMSE, 63] was used as a measure of general cognitive state. The detailed demographic information and lesion locations of each patient are presented in S1 Table.

Thirty-five healthy controls (21 males) matched to stroke patients by age, gender, and years of education participated in the study, with the following inclusion criteria: right-handed; normal vision or corrected vision; no contraindications for MRI; no history of neurological or psychiatric disorders; and intact color perception on the color vision examination plates test.

Note that one patient (ID: 018) had a premorbid distance vision problem (not improved with correction) and thus participated in the auditory version of the task fMRI experiment, where the stimuli were auditory words (names of fruits/vegetables) with otherwise the same presentation parameters as the visual version. In addition, while our patient inclusion criteria required first-time strokes, two patients' (ID: 003, 007) scans revealed potential old lesions. To ensure maximum lesion coverage, these three patients were included in the study and were further excluded in the control analyses.

## Procedures

**Task-fMRI experiment.** An object color judgment task-fMRI experiment was conducted to obtain object color neural representations in stroke patients and healthy controls. To minimize the potential confounding effect of task difficulty on the neural representations and measure meaningful neural activities, we tailored the task so that the patients could perform this task in the scanner with ease. We deliberately selected items that were familiar to the patients, adopted a simple yes/no judgment task, and performed the following procedure before the fMRI scanning to ensure that the participants recognized the grayscale stimuli of fruits and vegetables. The participants were first asked to name each stimulus displayed at the center of the computer screen. If a patient had a naming problem, he/she was given a word-to-picture matching session, where an auditory fruit/vegetable word was presented, and the patient needed to select the correct image out of the whole stimuli set. For the objects that the participants failed to name or match, they were provided with correct answers to ensure object recognition.

Grayscale images of 16 common fruits and vegetables (400 × 400 pixels; visual angle of 6.22 × 6.22 degrees) were used as stimuli. The images were matched for mean luminance separately for the foregrounds and backgrounds using the SHINE toolbox [64]. The number of pixels was comparable across the pictures (range: 29,889−31,866 pixels, mean = 30895.25, SD = 674.15). The visual stimuli were displayed using a Visual & Audio Stimulation System (Sinorad SA-9939) on a screen (89 cm width, 50 cm height; 171 cm from the participant's eyes). The participants viewed the display through an angled mirror (45°) attached to the headcoil. During scanning, the participants viewed the stimuli and verified whether the typical color of the skin of the fruit/vegetable was red (Fig 2A). The participants pressed a button with their right index finger to give a "yes" response and pressed another button with their right middle finger for a "no" response; stroke patients with lesions affecting the left hemisphere responded with their left hand.

All participants performed 4 runs (400 s per run) of slow-event-related task fMRI scanning, except for one patient (ID: 002) who completed three runs. Each run consisted of 32 trials, consisting of a 1-s-long stimulus period and an 11-s-long fixation period. Each stimulus was presented twice within each run. The order of the 32 trials was pseudorandomized, with the restriction that no two consecutive trials were identical. Each run began with a 12 s fixation period and ended with a 4 s fixation period. Independent pseudo-randomizations were created for each run for each participant. The experimental procedure was conducted using the E-prime 2 (Psychology Software Tools, Pittsburgh, PA, USA).

Individual behavioral RDMs of color, shape, and semantics were obtained for the 16 stimuli used in the fMRI task. The participants were asked to rate the similarity of these objects in terms of color, shape, and semantic relatedness (1: most similar, 7: most dissimilar), in a pairwise fashion (i.e., 120 pairs), respectively. Pairs were presented in pseudorandom order, with no time limit, and independent pseudo-randomizations were created for each feature rating. Before each rating, the participants practiced with 10 pairs of novel but familiar fruits and vegetables with similar and dissimilar examples

to understand the rating instructions. Individual RDMs were used for RSA (see below). Because we deliberately ensured that the items used in the fMRI experiment were familiar to the patients, the obtained group-averaged RDMs of the controls and patients were highly correlated (color: rho = 0.92, $p < 0.001$; shape: rho = 0.95, $p < 0.001$; semantics: rho = 0.94, $p < 0.001$).

Two items (chili and watermelon) were excluded from the analysis, because even healthy control participants generated highly variable colors for chili (red, green, yellow; all correct), and were uncertain about the color of watermelon referred to the skin (green) or the inside (red). Fourteen items were included in all the analyses below.

**Neuropsychological tests.** The participants were administered a series of neuropsychological tests outside the scanner to evaluate their object color behavioral task performances.

*Object color knowledge tasks*: Object color behavior was assessed on two tasks that varied in the modalities of input: a grayscale picture-color word matching task (verbal) and an object color true/false judgment (non-verbal) task. In the *grayscale picture-color word matching task*, participants were asked to judge whether the color word they heard matched the typical color of the vegetable/fruit in the grayscale picture. In the *object color true/false judgment*, a picture of a vegetable/fruit with typical or atypical colors appeared in the center of the screen, and participants were asked to judge whether the vegetable/fruit was typically colored. In both tasks, the same 30 familiar vegetables and fruits that appeared in different exemplars were used as stimuli. Each item appeared twice, once with the typical color or color word and once with the atypical one. The participants were asked to press the left mouse button for "yes" and the right mouse button for "no". The presentation order of the items was pseudorandom, and the same items did not appear consecutively, nor did the same responses appear consecutively three times. Each trial had a response deadline of 60 s. Responses were scored 1 if correct and 0 if wrong or if no response was given within the time deadline.

*Peripheral control tasks of object and color recognition*: To assess the (relative) neural specificity related to object color behaviors, we also carried out a word-picture matching task and a color patch matching task, which involved the two cognitive processes most relevant to object color processing: object recognition and color recognition. For the word-picture matching task, the stimuli included 20 familiar vegetables and fruits. In each trial, a chromatic picture of the vegetable/fruit appeared in the center of the screen, and an auditory vegetable/fruit word was played simultaneously. The participants were asked to judge whether the words matched the picture. Each item was presented twice, once in a matching condition and once in a non-matching condition. The stimulus presentation settings of this task were consistent with those of the object color knowledge tasks. In the color patch matching task, participants were asked to match the color word they heard to the 11 visually presented color patches (orange, green, gray, black, yellow, pink, purple, white, blue, brown, and red). The order of the color patches in each trial was pseudorandom, and each trial was presented without a time limit.

Tasks with grayscale pictures always preceded those with colored pictures to mitigate task-related priming. Each participant was tested in multiple sessions in a quiet room. Each session lasted for 2 h with pauses for rest upon request. Stimulus presentation was controlled by E-prime 1 (Psychology Software Tools, Pittsburgh, PA, USA).

One healthy control from the task fMRI participant cohort did not participate in the object color knowledge and control tasks; three healthy controls and one stroke patient did not participate in the color patch matching task; the patient with the visual problem (ID: 018) completed the auditory version of pairwise similarity ratings and the visual versions of other tasks (with corrected vision), but did not participate in the color patch matching task. In addition to the task fMRI cohort, another group of 17 healthy controls took part in the object color knowledge tasks and the word-picture matching task; one of them further finished the color patch matching task.

## Image acquisition

All functional and structural MRI data were collected using a 3T Siemens Magnetom Skyra scanner with a 32-channel radio frequency (RF) head coil, at the Department of Magnetic Resonance Imaging, First Hospital of Shanxi Medical

University. Foam padding was placed inside the RF head coil to minimize head movement. The scans included task-state fMRI, HARDI, high-resolution 3D T1-weighted images, 3D T2-weighted images, and 3D fluid-attenuated inversion recovery (FLAIR) T2-weighted images.

A multi-band echo-planar imaging (EPI) sequence was used for the fMRI and HARDI images. The acquisition parameters for fMRI were as follows: slice planes scanned along the rectal gyrus, phase encoding direction from posterior to anterior, repetition time (TR) = 2,000 ms, echo time (TE) = 30 ms, flip angle = 90°, axial scanning, field of view (FOV) = 180 mm × 180 mm, matrix size = 72 × 72, slice thickness = 2.5 mm, voxel size = $2.5 \times 2.5 \times 2.5$ mm$^3$, and multi-band factor = 2. The acquisition parameters for HARDI were as follows: diffusion gradient $b$ values set to 0, 1,000, and 2,000 s/mm$^2$, b0 repeated 10 times, 64 directions each for b1000 and b2000, TR = 3,000 ms, TE = 100 ms, flip angle = 90°, axial scanning, FOV = 224 mm × 224 mm, matrix size = 112 × 112, slice thickness = 2 mm, voxel size = $2 \times 2 \times 2$ mm$^3$, and multi-band factor = 2.

To maximize image resolution and increase the accuracy of lesion delineation, the T1-weighted, T2-weighted, and FLAIR T2-weighted images of this project all underwent 3D thin-layer scanning. The 3D T1-weighted images were acquired using a magnetization-prepared rapid gradient-echo sequence, with the following acquisition parameters: TR = 2,530 ms, TE = 2.88 ms, flip angle = 7°, sagittal scanning, FOV = 224 mm × 256 mm, matrix size = 224 × 256, interpolated to 448 × 512, slice thickness = 1 mm, and voxel size = $0.5 \times 0.5 \times 1$ mm$^3$. For the 3D T2-weighted images, a Sampling Perfection with Application optimized Contrast using different flip angle Evolution (SPACE) sequence was utilized, and the acquisition parameters were as follows: TR = 3,200 ms, TE = 408 ms, flip angle = 120°, sagittal scanning, FOV = 230 mm × 230 mm, matrix size = 256 × 256, slice thickness = 0.9 mm, and voxel size = $0.9 \times 0.9 \times 0.9$ mm$^3$. A SPACE Inversion Recovery sequence was used for the 3D FLAIR T2-weighted images, and the following acquisition parameters were used: TR = 5,000 ms, TE = 394 ms, flip angle = 120°, sagittal scanning, FOV = 250 mm × 250 mm, matrix size = 256 × 256, interpolated to 512 × 512, slice thickness = 1 mm, and voxel size = $0.5 \times 0.5 \times 1$ mm$^3$.

### Imaging data preprocessing

**fMRI data.** Analyses were conducted using SPM12 (RRID:SCR_007037; https://www.fil.ion.ucl.ac.uk/spm12/), based on MATLAB (Mathworks; RRID: SCR_001622). For each task run, the first six images were discarded to mitigate the effects of scanner instability at the onset of scanning. The remaining images underwent the following procedures: (1) temporal slice timing correction to adjust for the differences in acquisition time between slices within a single whole-brain scan; (2) realignment of each image frame to the first frame of the sequence to reduce the impact of head motion; (3) registration of EPI images to 3D T1 images; (4) normalization of 3D T1 images to MNI space using the unified segmentation algorithm [65]; (5) spatial normalization of EPI images using the normalization parameters obtained in the previous step, and resampling to a voxel size of $2 \times 2 \times 2$ mm$^3$. Two healthy controls were excluded from the fMRI analysis: one self-reported an uncomfortable feeling in the head during scanning, and the other exhibited excessive head motion during the scans (> 2.5 mm/2.5°). Four healthy controls and four stroke patients showed excessive head motion in 1 or 2 runs (healthy controls: > 2.5 mm/2.5°, stroke patients: > 3 mm/3°), and we analyzed the remaining runs of fMRI data for these participants.

**HARDI data.** Preprocessing was performed using the Functional Magnetic Resonance Imaging of the Brain (FMRIB) Analysis Group Software Library (FSL, version 6.07; Oxford University, United Kingdom), which included the following procedures: (1) Eddycorrect, which corrected for eddy current distortions and head movement; (2) BET, which was used for skull removal; (3) DTIFIT, which was used for building diffusion tensor models and calculating the FA maps. We subsequently registered the FA images to the T1 images in the native space using the FMRIB's linear image registration tool (FLIRT), and then used the T1-to-MNI transformation warp image to register the FA images to the Montreal Neurological Institute (MNI) space, with a target voxel size of $2 \times 2 \times 2$ mm$^3$. The warp image was obtained from normalizing 3D T1 images to MNI space with FLIRT for linear rigid affine transformations, and FMRIB's non-linear image registration tool (FNIRT) for non-linear transformations. The derived transformation parameters were then inverted

and used to warp the region of interest (ROI) from the MNI space to the native diffusion space using nearest-neighbor interpolation.

**Lesion mask drawing and normalization.** Lesions were manually drawn by a well-trained radiology resident (Y.H.) and reviewed by an experienced radiologist (Y.L.), using the ITK-SNAP software ([www.itksnap.org](www.itksnap.org)). The specific delineation and spatial normalization protocol included: (1) coregistering and resampling T2-weighted and FLAIR T2-weighted images to the native space of 3D T1-weighted images; (2) visually referring to the corresponding T1-weighted, and T2-weighted images, drawing on the axial plane of FLAIR T2-weighted images slice by slice, with gliosis included within the lesion outline; (3) employing the normalization parameters (obtained in step 4 of the fMRI data's preprocessing) to register the lesion images from native to MNI space, with a voxel size of $1 \times 1 \times 1\,mm^3$; and (4) applying a 3 mm smoothing kernel to the lesion images in standard space to obtain the patient-specific brain injury map (with 1 indicating injury and 0 indicating no injury). All processes and resulting normalized images were further inspected by an experienced clinical investigator (B.L.) for quality control.

## Data analysis

**Defining VOTC masks of interest in healthy controls.** *GLM*: For each participant, general linear modeling was conducted using preprocessed fMRI data. The general linear model (GLM) for each run included 16 regressors, each corresponding to one of the fruits or vegetables. The button presses and the six head motion parameters from each run were incorporated as regressors of no interest. Each regressor was convolved with the canonical SPM hemodynamic response function and high-pass filtered with a cutoff set at 128 s. The resulting *t* value maps for each vegetable/fruit compared with the baseline were obtained for subsequent multivariate pattern analysis.

*Object color RSA*: Searchlight RSA analyses were carried out with individual participants [66,67]. The neural RDM was constructed using Pearson's correlation distance of the activation patterns (*t* value maps) within a sphere (radius = 6 mm) centered on each GM voxel across 14 fruits and vegetables pairwise. The $14 \times 14$ neural RDM was then correlated with individuals' object color similarity RDM using partial Spearman's rank correlation, while controlling for low-level visual RDM (i.e., gist RDM), shape RDM, and general semantic RDM, to identify regions specifically encoding object color knowledge. The gist RDM measures the dissimilarity of low-dimensional visual representations across images, on the basis of spectral and coarsely localized information [68]. The resulting individual r maps were Fisher-Z transformed.

*Probability maps*: To account for inter-participant variability, the VOTC functional mask representing object color knowledge was localized using probability maps across healthy controls [45,69]. For each healthy control, the obtained RSA map specific to object color knowledge was masked with a functionally defined, VOTC-color-perception mask (defined in Wang and colleagues [22], by contrasting chromatic stimuli to grayscale stimuli in a color perceptual localizer in 14 healthy controls). In each masked image, the top 300 selected voxels (i.e., voxels with the highest Fisher-Z transformed *r* values) were binarized to construct the individual-level mask. Probability maps were created by summing the individual-level masks and dividing them by the number of healthy controls. Voxels that had a probability greater than 0.25 were retained as a group-level functional mask (i.e., VOTC-color-knowledge mask). To confirm the stability of the results produced by these arbitrary thresholds, we conducted validation analyses, using functional masks with various combinations of individual-level thresholds (top n voxels: 200, 250, 300, 350, 400, 450, and 500) and group-level thresholds (probability across healthy controls: 0.15, 0.2, 0.25, 0.3, and 0.35).

**Defining WM masks of interest in healthy controls.** In this phase of the study, we mapped the WM connections between the left language and visual perceptual systems, as well as the control connections, in healthy controls through quantitative analysis utilizing probabilistic tractography. Probabilistic tractography, an algorithm that investigates the probability distributions of fiber orientations at each voxel in the brain, offers the advantages of explicitly representing uncertainty [70] and better delineating crossing fibers in the brain [71,72]. Particularly for datasets with multiple *b*-values (e.g., the HARDI data we collected), the application of probabilistic tractography with a gamma distribution may prove

advantageous for combining multiple shells, thereby increasing the accuracy of fiber orientation estimates in WM and areas adjacent to the cortex [73].

*ROI selection*: The ROIs included a set of left language-specific regions, defined by Fedorenko and colleagues (contrasting intact sentences to nonword lists in 220 healthy subjects), the functionally defined VOTC-color-perception mask (defined in Wang and colleagues [22], by contrasting chromatic stimuli to grayscale stimuli in a color perceptual localizer in 14 healthy subjects), and the early visual cortex (the OP (48#) in the Harvard-Oxford Atlas, probability > 0.2). The right hemisphere homologs of the left-hemisphere language regions, as defined by Fedorenko and colleagues, were also included to investigate the lateralization of the observed effects. Given that different subdivisions of the ATL may serve different aspects of semantic representation, especially along the dorso-ventral axis ([21,22,43]; see Bi [24], Lambon Ralph and colleagues [5] for review), we examined the specificity of the LdlATL by further including different ATL subregions as ROIs. These subregions were parcellated either by functional activation connectivity profiling [43] or anatomical profiling (i.e., Harvard-Oxford Atlas (probability > 0.2)). For the functional activation connectivity profiling, the ATL was parcellated into four subregions (superior dorsal, inferior dorsal, lateral, and ventromedial) based on coactivation clustering, and the two dorsal subregions in that study were combined here into a single dorsal subregion. For the anatomical profiling, according to the Harvard-Oxford Atlas (probability > 0.2), the ATL subregions included the temporal pole (8#), the anterior superior temporal gyrus (9#), the anterior middle temporal gyrus (11#), the anterior inferior temporal gyrus (14#), the anterior parahippocampal gyrus (34#), and the anterior temporal fusiform cortex (37#). These ROIs were transformed to the diffusion native space for each participant via the inverse of the linear and nonlinear transformations as previously described.

*Probabilistic tractography*: Probabilistic tractography was carried out using healthy controls' HARDI data with FMRIB's Diffusion Toolbox (FDT v5.0, http://fsl.fmrib.ox.ac.uk/fsl/fslwiki/FDT) in FSL. We employed the ROI-to-ROI approach, in which tractography was implemented between each pair of ROIs for the following tracts: (1) the main tracts of interest—the connection between the VOTC and the left language regions; (2) the tract involved in lower-level visual perception—the connection between the VOTC and the early visual cortex; (3) the right hemisphere homologs of the main tracts of interest—the connections between the VOTC and the right homologs of the left language regions; and (4) the connections between the VOTC and the ATL subregions. The detailed processes are as follows.

Tractography was estimated using FSL's BEDPOSTX with default parameters to model WM fiber orientations and crossing fibers, where the diffusion coefficient was modeled using a gamma distribution. Fiber tracking was initiated in both directions (from seed to target and vice versa) in the diffusion space, and 5,000 streamlines were drawn from each voxel in the ROI using Probtrackx 2.0. A binarized cerebrospinal fluid (CSF) mask (generated by setting the probability threshold > 0.75 for the CSF probability mask in SPM12) was set as an exclusion mask for the analyses. The resulting images containing the output connectivity distribution were normalized to MNI space and standardized using the maximum voxel intensity of each image, resulting in a standardized image with voxel values ranging from 0 to 1. We then applied an individual-level threshold for reducing false-positive fiber tracks combined with a group-level threshold for the fiber tracking success rate [74,75]. At the individual level, the standardized path images were thresholded at 0.1 to remove voxels with a low connectivity probability. The resulting maps were then binarized for each individual and summed across individuals. Next, at the group level, fiber projections that existed in more than 50% of the individuals within the explicit WM mask (probability > 0.4 in the WM probability map in SPM12) were retained for subsequent analysis. The threshold was selected arbitrarily, which is a common practice in tracking studies for reporting group results in general anatomical assessments. We further conducted validation tests, using WM connection masks with various combinations of individual-level thresholds (connectivity probability: 0.05, 0.1, 0.15, 0.2) and group-level thresholds (probability: 0.35, 0.4, 0.45, 0.5, 0.55, 0.6), with or without the explicit WM mask. To verify the tracts we reconstructed, we overlaid each tract onto the 20 major tracts in the JHU WM tractography atlas (thresholded at 25%; http://fsl.fmrib.ox.ac.uk/fsl/fslwiki/Atlases) in FSL. The overlap percentage was computed by dividing the number of common voxels by the total number of voxels in the template tract.

**VOTC object color neural representation in patients: Effects of the WM tract integrity and specificity tests.** To investigate whether the lesions to the tracts connecting the left language regions and the VOTC resulted in impaired VOTC object color neural representation, we conducted partial Spearman's rank correlation analyses between the WM integrity of each tract and the strength of the VOTC object color neural representation across patients, with each patient's total lesion volume as a covariate. For WM integrity, we computed the mean FA value within each tract. For the strength of the VOTC object color neural representation, we extracted the mean values from the object color RSA results within the VOTC color-knowledge mask (defined in healthy controls above). A false discovery rate (FDR) with a threshold of $q < 0.05$ was adopted to correct for comparisons of multiple tracts. Given the potential overlaps among the masks of the language-VOTC tracts, we carried out a specificity test by excluding voxels that overlapped with other tracts and considering only the mean FAs of the tract-specific voxels and then computed Spearman's rank correlations.

To investigate whether the observed effects of interest were influenced by potential alternative neuroanatomical and/or cognitive variables, in addition to controlling for total lesion volume, we conducted a series of control analyses: (1) *GM lesions*. To examine whether the WM effects are inherited from relevant GM lesions, we performed partial correlations, including the lesion percentage in the GM regions at both ends of this connection as covariates. The GM areas were identical to the seed regions used in probabilistic tractography, and lesion percentages in these GM regions were calculated as the number of voxels with lesions divided by the total number of voxels in the region. (2) *Visual perception pathway.* Considering that the VOTC functionality may be affected by lesions in the WM tracts connecting to the early visual cortex, we thus further added the mean FA values of these WM connections as covariates. (3) *Patient etiology.* Data from three patients (ID: 003, 007, 018) warrant scrutiny, as described in the subsection "Participants". To eliminate the potential influence of these special patients, we performed correlation analyses after their exclusion. We further included the time post-onset (month) as a covariate along with total lesion volume given the substantial variation in this variable (ranging from 3 to 252 months).

**Object color behavior in patients: Underlying neural correlates and specificity tests.** *Behavioral data analysis*: For the two object color knowledge tasks (grayscale picture-color word matching, object color true/false judgment) and peripheral control tasks (word-picture matching, color patch matching), accuracy scores were analyzed. Given the variations along the demographic variables (i.e., age, gender, and years of education), we employed a statistical method to adjust for these variables. The Bayesian test for a Deficit allowing for Covariates [41] was used to compare a patient's performance with that of controls and to obtain the point estimates of effect sizes for the difference between a patient and controls (controlling for the influence of three demographic variables) as standardized scores. Cases with one-tailed $p < 0.05$ were considered to have (potential) deficits.

*Correlation analyses*: In this section, we examined how the VOTC and related connections support object color behavior. Hence, we performed partial Spearman's rank correlation analyses between the strength of the VOTC object color neural representation or the mean FA value of the observed tracts and the neuropsychological test performance across patients, controlling for total lesion volume. As object color behaviors were assessed in two tasks that varied in terms of input modalities, we computed a composite score by averaging the effect size estimate of accuracy from the two tasks (controlling for the three demographic variables and considering the variation in healthy controls). When correlated with neural measures, the composite score was considered and reported in the main text. The results of each individual task are shown in S4–S6 Tables. Differences between the correlations for the two tasks were compared using Hotelling's *t* test (the FZT computator, http://psych.unl.edu/psycrs/statpage/regression.html), with results reported in S6 Table.

*Specificity tests*: To investigate whether the observed results for WM tract integrity in object color behavior were influenced by various types of potentially confounding variables, we also conducted a series of control analyses by including lesions to GM regions, visual perception pathway integrity, and patient etiology as covariates. Importantly, to assess whether the observed tracts of interest were specifically associated with object color knowledge relative to other cognitive processes, we included the standardized scores of the word-picture matching task and the color patch matching task as covariates in addition to total lesion volume.

## Supporting information

**S1 Fig. Object-color RSA effects in the whole-brain searchlight analysis in the healthy controls (voxel-level $p < 0.001$, one-tailed). (A)** Raw object-color representation (correlating neural RDMs with the behavioral object color similarity RDM) peaked in the right lingual gyrus (peak MNI xyz: 16, −88, −8; peak $t = 4.01$; 35 voxels). **(B)** Unique object-color representation, after controlling for low-level visual RDM, shape RDM, and general semantic RDM, showed similar anatomical locations (peak MNI xyz: 16, −90, −8; peak $t = 3.71$; 19 voxels). Brain results were visualized using BrainNet Viewer (version 1.7; https://www.nitrc.org/projects/bnv/; RRID: SCR_009446).
(PDF)

**S2 Fig. Group-level white matter connection probability map between the bilateral VOTC (green) and the dlATL (blue) at different individual thresholds in healthy controls.** The red color scale (from 0 to 1) indicates the group-level probability of a voxel belonging to a white-matter connection. **(A)** Group-level probability maps of white-matter connectivity between the bilateral VOTC and the left dlATL (top) and right dlATL (bottom), respectively, at the individual threshold of 0.1 (the threshold used in the main analysis). **(B)** Group-level probability map between the bilateral VOTC and the left dlATL at an individual threshold of 0.001. Brain results were visualized using MRIcron (version 1.0.20190902; https://www.nitrc.org/projects/mricron; RRID: SCR_002403). *Abbreviations: VOTC, ventral occipitotemporal cortex; dlATL, dorsolateral anterior temporal lobe.*
(PDF)

**S3 Fig. Effects of the white-matter tracts connecting the VOTC with the other five language parcels on VOTC neural representations and object color behaviors.** The first row illustrates the seed regions for probabilistic tractography and the corresponding reconstructed white-matter tracts in healthy controls: VOTC-LpMTG, VOTC-LAG, VOTC-LIFGorb, VOTC-LIFG, and VOTC-LMFG. The second row shows the scatter plots between the mean FA values of these tracts and the VOTC object-color neural representation (Fisher-Z transformed $r$ values). The third row shows the scatter plots between these tracts and object color behavior (the composite score across the verbal and non-verbal object color tasks). Similar to Fig 3, the behavioral outlier patient was removed from the scatter plots with object color behaviors and the reported partial rho and $p$-values were based on data from all the patients. The data underlying this figure are available in S1 Data. Brain imaging results were visualized using MRIcroGL (version 1.2.20210317; https://www.nitrc.org/projects/mricrogl). *Abbreviations: VOTC, ventral occipitotemporal cortex; L, left; pMTG, posterior middle temporal gyrus; AG, angular gyrus; IFGorb, inferior frontal gyrus, orbital part; MFG, middle frontal gyrus; FA, fractional anisotropy.*
(PDF)

**S4 Fig. Proportion of subjects providing correct responses in the color patch matching task.** The overall proportion of "orange" and "brown" words in the patient group was less than 100% because two patients thought that the two color words did not match any color patches presented in the image. The data underlying this figure are available in S1 Data.
(PDF)

**S5 Fig. Effects of the tracts connecting the VOTC and six ATL subregions on VOTC neural representations and object color behaviors.** The six ATL subregions were taken from the HOA template (see Materials and methods). The first row displays the seed regions for probabilistic tractography and the corresponding reconstructed white-matter tracts: VOTC-LTP, VOTC-LASTG, VOTC-LAMTG, VOTC-LAITG, VOTC-LATFC, and VOTC-LAPG. The second row illustrates the correlations between these six tracts and VOTC object color neural representation. The third row shows the correlations between these six tracts and object color behavior (the composite score across the verbal and non-verbal object color tasks). The white-matter connections between the three ventral subregions (LAITG, LATFC, and LAPG) and the VOTC were disrupted and had fewer than 100 voxels under the thresholds used in the main analysis (individual-level

threshold of 0.1 and a group-level threshold of 0.5, masked by the SPM12 white-matter map with probability > 0.4). The tracts shown here were obtained using more lenient thresholds (individual-level threshold of 0.1 and group-level threshold of 0.4, without the explicit white-matter mask). The correlations reported in the scatter plots were based on the tracts reconstructed using the more lenient thresholds, and similar correlations were observed with the tracts under the default thresholds. Similar to Fig 3, the behavioral outlier patient was removed from the scatter plots with object color behaviors and the reported partial rho and *p*-values were based on data from all the patients. The data underlying this figure are available in S1 Data. Brain imaging results were visualized using MRIcroGL (version 1.2.20210317; https://www.nitrc.org/projects/mricrogl). *Abbreviations: VOTC, ventral occipitotemporal cortex; L, left; TP, temporal pole; ASTG, anterior superior temporal gyrus; AMTG, anterior middle temporal gyrus; AITG, anterior inferior temporal gyrus; ATFC, anterior temporal fusiform cortex; APG, anterior parahippocampal gyrus; FA, fractional anisotropy; HOA, Harvard-Oxford Atlas.*
(PDF)

**S6 Fig. A case of damage to the VOTC-LdlATL tract.** Lesion profile. The three sagittal slice images (in the top left panel) show T1-weighted, T2-weighted, and FLAIR T2-weighted images of the same slice in the native space. The cortical map and the slice images (in the bottom panel) show the overlap of the patient's lesions with the VOTC-LdlATL tract in the MNI space, illustrating that the patient's VOTC-LdlATL tract was severely lesioned. Two types of lesions were considered here: manually drawn lesion masks and voxels with significantly lower FA values. The low-FA voxels, defined as voxels with FA values significantly lower than those of healthy controls according to the single-case modified *t* test [76]; *p* < 0.05, one-tailed], was to quantify invisible lesions. The top right panel shows the extent of overlap in the proportion of lesions within the VOTC-LdlATL tract. The manually drawn lesion mask overlapped with 35.5% of the VOTC-LdlATL tract, consisting of 26.3% with significantly lower FA values (blue area) and 9.2% (yellow area) whose FA values were in the normal range. 22.7% of the tract (green area) was found to have lower FA values and was outside the manually drawn lesion mask. The red area shows the portion of the VOTC-LdlATL tract that was not affected in the patient; the gray area shows the lesion outside the VOTC-LdlATL tract. Brain imaging results were visualized using BrainNet Viewer (version 1.7; https://www.nitrc.org/projects/bnv/; RRID: SCR_009446), or MRIcron (version 1.0.20190902; https://www.nitrc.org/projects/mricron; RRID: SCR_002403). *Abbreviations: VOTC, ventral occipitotemporal cortex; L, left; dlATL, dorsolateral anterior temporal lobe; FA, fractional anisotropy.*
(PDF)

**S1 Table. Background information of the 33 stroke patients.**
(DOCX)

**S2 Table. Raw accuracy and standardized scores of each stroke patient in the four neuropsychological tests.**
(DOCX)

**S3 Table. The anatomical properties of the white-matter tracts connecting the VOTC and the left language regions in healthy controls.**
(DOCX)

**S4 Table. Validation of the VOTC object color neural representation effects (correlations with the VOTC-LdlATL white-matter connection and object color behaviors), using VOTC-color-knowledge masks with different individual-level and group-level thresholds.** The data underlying this table are available in S1 Data.
(DOCX)

**S5 Table. Validation of the effects of the VOTC-LdlATL white-matter connection (with different individual-level and group-level thresholds, with or without the explicit white-matter mask) on VOTC object color neural representation and object color behaviors.** The data underlying this table are available in S1 Data.
(DOCX)

**S6 Table. Partial correlation coefficients between the VOTC-LdlATL tract integrity (mean FA value) and performances on the grayscale picture-color word matching (verbal) and object color true/false judgment (non-verbal) tasks, along with between-task comparisons of the correlations using Hotelling's *t* test, controlling for potential confounding factors.** The data underlying this table are available in S1 Data.
(DOCX)

**S7 Table. Summary of previous neuroimaging findings on object color knowledge (not exhaustive).**
(XLSX)

**S1 Text. A case of damage to the VOTC-LdlATL tract.**
(DOCX)

**S1 Data. The data necessary to reproduce** Figs 2–4, **S3**–**S5** and Tables 1, **S4**–**S6**.
(XLSX)

## Acknowledgments

We thank Weiwei Men for help with imaging methodology, Yuxing Fang for help with tractography methodology, and all members of the Department of Radiology at the First Hospital of Shanxi Medical University for data collection. We are also grateful to all research participants.

## Author contributions

**Conceptualization:** Bo Liu, Yanchao Bi.

**Data curation:** Bo Liu, Yan Li, Yang Han.

**Formal analysis:** Bo Liu, Jiahui Lu.

**Funding acquisition:** Xiaosha Wang, Xiaoying Wang, Yanchao Bi.

**Investigation:** Bo Liu, Xiaosha Wang, Xiaoying Wang, Yanchao Bi.

**Methodology:** Bo Liu, Xiaosha Wang.

**Project administration:** Xiaochun Wang, Yanchao Bi.

**Resources:** Hui Zhang, Xiaochun Wang.

**Software:** Bo Liu.

**Supervision:** Hui Zhang, Xiaochun Wang, Yanchao Bi.

**Validation:** Bo Liu, Xiaosha Wang, Jiahui Lu.

**Visualization:** Bo Liu, Xiaosha Wang, Yanchao Bi.

**Writing – original draft:** Bo Liu, Xiaosha Wang, Yanchao Bi.

**Writing – review & editing:** Bo Liu, Xiaosha Wang, Yanchao Bi.

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
