## [Editor Report · Decision Letter 0]

24 Oct 2024

Dear Dr Bi, 

Thank you for submitting your manuscript entitled "Visual cortex object representation in the human brain needs connection with the language systems" for consideration as a Research Article by PLOS Biology.

Your manuscript has now been evaluated by the PLOS Biology editorial staff as well as by an academic editor with relevant expertise and I am writing to let you know that we would like to send your submission out for external peer review.

Once your full submission is complete, your paper will undergo a series of checks in preparation for peer review. After your manuscript has passed the checks it will be sent out for review. To provide the metadata for your submission, please Login to Editorial Manager (https://www.editorialmanager.com/pbiology) within two working days, i.e. by Oct 26 2024 11:59PM.

Kind regards,

Christian

Christian Schnell, PhD

Senior Editor

PLOS Biology

cschnell@plos.org

---

## [Decision Letter · Decision Letter 1]

13 Dec 2024

Dear Dr Bi,

Thank you for your patience while your manuscript " Visual cortex object representation in the human brain needs connection with the language systems " was peer-reviewed at PLOS Biology. It has now been evaluated by the PLOS Biology editors, an Academic Editor with relevant expertise, and by several independent reviewers. 

In light of the reviews, which you will find at the end of this email, we would like to invite you to revise the work to thoroughly address the reviewers' reports.

As you will see below, the reviewers agree that the study provides potentially important insides. However, all reviewers have concerns regarding the presentation and the logical flow. Other concerns are inconsistencies with the existing literature, lack of methodological details, and the unclear role of both hemispheres.

Given the extent of revision needed, we cannot make a decision about publication until we have seen the revised manuscript and your response to the reviewers' comments. Your revised manuscript is likely to be sent for further evaluation by all or a subset of the reviewers.

**IMPORTANT - SUBMITTING YOUR REVISION**

*Re-submission Checklist*

*Published Peer Review*

*PLOS Data Policy*

*Blot and Gel Data Policy*

Sincerely,

Christian

Christian Schnell, PhD

Senior Editor

PLOS Biology

cschnell@plos.org

REVIEWS:

Reviewer #1: Summary. This study examined the necessary role of language regions in the neural representation of semantic information, specifically for knowledge of object color. Thus, the research addressed fundamental questions about the extent to which semantic knowledge representations are sensory in nature or have language-based aspects. Using representational similarity analyses (RSA) of fMRI data from healthy individuals, regions of the ventral-occipital cortex with the greatest involvement in the representation of object color were identified and used to create a color-knowledge mask. Probabilistic tractography of diffusion imaging was successful in identifying white matter tracts connecting this color-knowledge mask to six language regions identified using a language localizer task contrasting the processing of sentences vs. lists of nonwords. 

33 patients with lesions from stroke that spared posterior regions were studied with the same behavioral and imaging protocols as for the healthy subjects. For the patients, the tract with the highest correlation between FA values and the mean RSA values in the VOTC mask had a termination in the left dorsal-lateral anterior temporal lobe (LdlATL), a region that had been implicated in color knowledge in the congenitally blind. The results suggest that input from the language region determines how well object color knowledge is represented in the VOTC. Also, behavioral scores on the color tasks were significantly correlated with the integrity of the tract linking VOTC to LdlATL but not significantly correlated with any other tract. Interestingly the behavioral scores were not related to RSA decoding in the VOTC. 

Critique. The results strongly implicate a role for language representations in semantic knowledge of object color. The research was carefully done, ruling out alternative explanations for the significant relations involving the LdlATL, such as damage to the gray matter regions at the termini. However, there are several issues that should be addressed before the paper is considered for publication.

1) More information is needed about the tractography. The identified language regions were all in the left hemisphere but the VOTC included bilateral regions. How are tracts identified that go from the right hemisphere portions of VOTC to the left hemisphere language regions? Presumably the tracts would have to include the corpus callosum or some other tract connecting the two hemispheres. Or perhaps only left hemisphere parts of the VOTC mask were used, which is suggested by the presentation of the analysis of homologous right hemisphere tracts in lines 338-340. 

2) What is to be made of the fact that there was no correlation between patient behavior and the RSA values for the VOTC? Given that the strength of the WM tracts relates to decoding of color in the VOTC, implying an influence of language on color knowledge, why wouldn't that degree of color knowledge relate to behavior?

3) While the authors point out that the language processing involved was minimal in the tasks with a verbal component, greater emphasis should be placed on the fact that the significant correlations observed for the stroke patients persisted when only looking at the semantic tasks that were purely nonverbal. 

Specific points.

Abstract. Something is said about possible implications of the findings for rehabilitation, but nothing is said in the paper about this topic.

Lines 139-155 This description was somewhat confusing. A color perception region was established and then apparently object color knowledge regions were only identified within that region. Is that known to be the case? Some references might be helpful.

Line 90-91 Is calcarine cortex a higher visual processing area than VOTC? Line 236-237 Same question for occipital pole.

Line 186 What kind of word cues were used? 

Other minor points where wording needs correction or further explication:

Lines 68-69 "origin of the potential modulation" is unclear

Lines 72 - "fully depriving sensory experience" perhaps s/b "resulting in full deprivation of sensory experiences"

Line 81 "with necessity and/or sufficiency" should be reworded.

Line 81-82 Something is missing in "does VOTC the neural representation know". Maybe s/b "does the neural representation in VOTC "know"

Line 85 "but inconclusive" s/b "but this evidence is inconclusive"

Line 89 "atrophy is not confomative" Conformative with what?

Line 437 Perhaps s/b "these results are in accord with previous results showing that these large WM bundles are Involved in supporting…"

Reviewer #2: This is a study on 33 stroke survivors and 35 age-matched controls. Participants performed fMRI experiments of object color knowledge: they were shown greyscale pictures of familiar fruits and vegetables. They were asked to judge whether their typical skin color of the fruit/vegetable was red. White matter tracts were studied using diffusion MRI. The main result (as far as I understand it) is that the structural integrity of white matter connections between high-level visual cortex and the left anterior temporal lobe was causally related to object-color knowledge.

This study contains potentially important results, but in its current form it's difficult or sometimes even impossible to follow. In my opinion, the paper should be rewritten in a much more streamlined and clarified version. The new version should follow a clear experimental logic; important results should be properly highlighted. For example, if, as it seems, the main topic of the study is the brain organization of object-color knowledge, why is there no mention of color in the title, and color only appears midway in the abstract? This is confusing.

Other points:

Abstract: I don't see any direct link between the study results and "embodied object knowledge neural representation."

The hemispheric side of lesion is obviously highly relevant in this study on the role of language in color knowledge, but it is not even clear from the outset how many of the 33 patients had left hemisphere lesions. 

Line 120. "typical lesion distribution for a stroke population." Indeed, most patients seem to have lesions in the middle cerebral artery territory, which is indeed the most frequent stroke localization. However, the most informative lesion sites to test the authors' hypotheses on VOTC are in the territory of the posterior cerebral artery. 

Line 90. "the object knowledge deficits might still be attributable to the VOTC's decreased structural connection between the left fusiform gyrus and calcarine rather than anterior integrity." But this would be highly unlikely because patients with cortical blindness from bilateral V1 lesions or with cerebral achromatopsia can show perfectly normal color knowledge (see, e.g., Bartolomeo, P., Liu, J., & Spagna, A. (2024). Colors in the mind's eye. _Cortex_, _170_, 26-31).

Line 152. "The group-level whole-brain analyses in the healthy controls revealed the right lingual gyrus as the top cluster representing object color knowledge." This result looks really implausible. It is simply false that this finding is, as the authors state, "in line with those reported in the literature" (line 154). On the contrary, abundant evidence points to the *left* VTC. See, eg, one of the studies cited in the present paper: "We observed increased activation during both perception and memory retrieval of chromatic compared to achromatic stimuli in overlapping areas of the *left* lingual gyrus" (Hsu et al, 2012). Another cited study (Wang et al 2020) discusses "object-color-knowledge representation effects in the *left* ATL." The right hemisphere result might be an artifact of co-variating for object shape RDM: the authors are testing for object colors, not abstract colors. Object-color knowledge probably requires the integration of object shape and color in the brain. Thus, controlling for object shape can lead to artifactual results. More generally, the authors should be much more careful when citing the existing literature. I didn't check all the other citations.

Line 187: Patients had similar performance accuracy (0.90) and SD (0.09) as controls (0.92 ± 0.06) on the color knowledge test. How can their lesion or WM pattern be informative on the neural bases of color knowledge?

Line 483. "Thirty-three stroke patients (25 males), without major posterior lesions". This exclusion criterion needs to be motivated. By the way, what does "posterior" mean here anatomically? Postrolandic?

Minor points

The English is often incorrect and should be reviewed. E.g., in the abstract: "in a group of patients suffered from stroke" should be "having suffered from" or something similar.

Line 89. What does "atrophy is not conformative" mean?

Reviewer #3: The question of the relationship between language and cognition is an old and perentially interesting one. Recent work by Fedorenko et al., has argued in favor for the modularity of language from the rest of cognition based on language localizer results. The study here is a fascinating counterargument, showing neural evidence in favor of linguistic modulation of ventral occipitotemporal cortex (VOTC) that has been previously linked to knowledge of object color (and perhaps visual knowledge more generally). 

This is a valuable addition to the literature.

I have two main questions/comments:

First, I wish the authors made it a bit clearer what they think the language regions are doing. How are they modulating VOTC? As a source of ideas — and I don't make a habit of citing myself in reviews — please see:

Lupyan, G. (2009). Extracommunicative Functions of Language: Verbal Interference Causes Selective Categorization Impairments. *Psychonomic Bulletin & Review*, *16*(4), 711-718. ~[https://doi.org/10.3758/PBR.16.4.711](https://doi.org/10.3758/PBR.16.4.711)~

and

Lupyan, G., Mirman, D., Hamilton, R. H., & Thompson-Schill, S. L. (2012). Categorization is modulated by transcranial direct current stimulation over left prefrontal cortex. *Cognition*, *124*(1), 36-49. ~[https://doi.org/10.1016/j.cognition.2012.04.002](https://doi.org/10.1016/j.cognition.2012.04.002)~

(as well as the citations to older literature, e.g., Cohen, Kelter, and other members of the Konztanz group — cited in the lit review)

Second, I am surprised at the variability of the performance of the control group (Fig 3A). Can you say something abotu what predicts the *control* group performance? I understand that the control group served as a baseline for identifying tractographic norms against which the patients were compared, but is there a way to use the tractography analyses in the contrl group to see if any differences in the connection between language regions and VOTC predict *their* performance? 

Other comments:

The description of the behavioral methods make it somewhat hard to connect the methods to the relevant figures, to the relevant results. For example, it's not clear what the numeric results are in the task shown in Fig. 2A (and it's not clear what it's called, making it difficult to find the relevant bit of methods). 

Please proofread for grammar - there are many places with incorrect number agreement (e.g., ) and other typos, e.g., "comes from patient studies, but inconclusive"

-Gary Lupyan

---

## [Decision Letter · Decision Letter 2]

19 Mar 2025

Dear Dr Bi,

Thank you for your patience while we considered your revised manuscript "Visual cortex object representation in the human brain needs connection with the language systems" for publication as a Research Article at PLOS Biology. This revised version of your manuscript has been evaluated by the PLOS Biology editors, the Academic Editor and the original reviewers.

Based on the reviews and on our Academic Editor's assessment of your revision, we are likely to accept this manuscript for publication, provided you satisfactorily address the remaining points raised by the reviewers and the following data and other policy-related requests:

* We would like to suggest a different title to improve its accessibility for our broad audience: "Object representation in the human visual cortex requires a connection with the language system"

* Please include information in the Methods section whether the study has been conducted according to the principles expressed in the Declaration of Helsinki.

* CODE POLICY

Per journal policy, if you have generated any custom code during the course of this investigation, please make it available without restrictions. Please ensure that the code is sufficiently well documented and reusable, and that your Data Statement in the Editorial Manager submission system accurately describes where your code can be found. [

* Please note that per journal policy, we do not allow the mention of "data not shown", "personal communication", "manuscript in preparation" or other references to data that is not publicly available or contained within this manuscript. Please either remove mention of these data or provide figures presenting the results and the data underlying the figure(s).

We expect to receive your revised manuscript within two weeks. 

*Published Peer Review History*

*Press*

Sincerely,

Christian

Christian Schnell, PhD

Senior Editor

cschnell@plos.org

PLOS Biology

Reviewer remarks:

Reviewer #1: The authors have done an excellent job in addressing my comments and those of the other reviewers. The paper will make a nice contribution to the literature.

Reviewer #2: The authors should be commended for revising their paper extensively, rendering it more readable. 

I'm still uncertain about the anatomy, particularly how lesions in the MCA territory could affect the white matter tracts identified as critical for color knowledge. Given that ILF and IFOF are supplied by the PCA, it remains unclear how damage within the MCA territory would disrupt these pathways. Perhaps a disconnectome analysis (see https://rdcu.be/ecVVX) could help clarify this issue.

The authors conclude the abstract by stating that their findings provide "novel anatomical and functional targets for the rehabilitation of knowledge loss." However, I find this claim and its elaboration in the discussion too vague. How exactly could knowledge loss be restored by targeting these regions? Unless the authors have something specific in mind, I would suggest removing the mention of rehabilitation.

Reviewer #3: The reviewers did a good job addressing my earlier concerns. In my estimation I think they also adequately addressed the (somewhat more critical) reviews of the other referees. My one remaining suggestion is to make the behavioral results a bit more salient, e.g., please add the means to the Fig 3A plots and make the control-patient inferential test more prominent in the text. Which contrasts show significant differences?

---

## [Editor Report · Decision Letter 3]

12 Apr 2025

Dear Dr Bi,

Thank you for the submission of your revised Research Article entitled "Object knowledge representation in the human visual cortex requires a connection with the language system" for publication in PLOS Biology. On behalf of my colleagues and the Academic Editor, Huan Luo, I am pleased to say that we can in principle accept your manuscript for publication, provided you address any remaining formatting and reporting issues. These will be detailed in an email you should receive within 2-3 business days from our colleagues in the journal operations team; no action is required from you until then. Please note that we will not be able to formally accept your manuscript and schedule it for publication until you have completed any requested changes.

PRESS

Sincerely, 

Ines

--

Ines Alvarez-Garcia, PhD

Senior Editor

PLOS Biology

on behalf of

Christian Schnell, PhD, 

Senior Editor

PLOS Biology

cschnell@plos.org